# Multi-label classification: do Hamming loss and subset accuracy really conflict with each other?

**Guoqiang Wu,    Jun Zhu**[*]
Dept. of Comp. Sci. & Tech., Institute for AI, BNRist Center
Tsinghua-Bosch Joint ML Center, THBI Lab, Tsinghua University, Beijing, 100084 China
Jiangsu Collaborative Innovation Center for Language Ability, Jiangsu Normal University, China
guoqiangwu90@gmail.com,   dcszj@mail.tsinghua.edu.cn

## Abstract

Various evaluation measures have been developed for multi-label classification, including Hamming Loss (HL), Subset Accuracy (SA) and Ranking Loss (RL). However, there is a gap between empirical results and the existing theories: 1) an algorithm often empirically performs well on some measure(s) while poorly on others, while a formal theoretical analysis is lacking; and 2) in small label space cases, the algorithms optimizing HL often have comparable or even better performance on the SA measure than those optimizing SA directly, while existing theoretical results show that SA and HL are conflicting measures. This paper provides an attempt to fill up this gap by analyzing the learning guarantees of the corresponding learning algorithms on both SA and HL measures. We show that when a learning algorithm optimizes HL with its surrogate loss, it enjoys an error bound for the HL measure independent of $c$ (the number of labels), while the bound for the SA measure depends on at most $O(c)$. On the other hand, when directly optimizing SA with its surrogate loss, it has learning guarantees that depend on $O(\sqrt{c})$ for both HL and SA measures. This explains the observation that when the label space is not large, optimizing HL with its surrogate loss can have promising performance for SA. We further show that our techniques are applicable to analyze the learning guarantees of algorithms on other measures, such as RL. Finally, the theoretical analyses are supported by experimental results.

## 1   Introduction

Multi-label classification (MLC) [1] is a fundamental task that deals with the learning problems where each instance might be associated with multiple labels simultaneously. It has enjoyed applications in a wide range of areas, such as text categorization [2], image annotation [3], etc. It is more challenging than the multi-class classification problem where only one label is assigned to each instance. Due to the complexity of MLC, various measures [4, 5] have been developed from diverse aspects to evaluate its performance, e.g., Hamming Loss (HL), Subset Accuracy (SA) and Ranking Loss (RL). To optimize one or a subset of these measures, plenty of algorithms [3, 6, 7, 8] have been proposed. For instance, Binary Relevance (BR) [3] aims to optimize HL while Rank-SVM [6] aims to optimize RL. For a comprehensive evaluation of different algorithms, it is a common practice to test their performance on various measures and a better algorithm is the one which performs well on most of the measures. However, it is commonly observed that an algorithm usually performs well on some measure(s) while poorly on others. Thus, it is important to theoretically understand such inconsistency to reveal the intrinsic relationships among the measures.

---

[*]corresponding author

Table 1: Summary of the main theoretical results in this paper.

| Bound w.r.t. / Algorithm | Hamming Loss | Subset Loss[1] | Ranking Loss | Proposed |
|---|---|---|---|---|
| Optimize Hamming Loss ($\mathcal{A}^h$) | $\hat{R}_S^h(f) + O(\sqrt{\frac{1}{n}})$ | $c\hat{R}_S^h(f) + O(\sqrt{\frac{c^2}{n}})$ | $c\hat{R}_S^h(f) + O(\sqrt{\frac{c^2}{n}})$ | [3] |
| Optimize Subset Loss ($\mathcal{A}^s$) | $\hat{R}_S^s(f) + O(\sqrt{\frac{c}{n}})$ | $\hat{R}_S^s(f) + O(\sqrt{\frac{c}{n}})$ | $\hat{R}_S^s(f) + O(\sqrt{\frac{c}{n}})$ | This paper |
| Optimize Ranking Loss ($\mathcal{A}^r$) | $c\hat{R}_S^r(f) + O(\sqrt{\frac{c^3}{n}})$ | $c^2\hat{R}_S^r(f) + O(\sqrt{\frac{c^5}{n}})$ | $\hat{R}_S^r(f) + O(\sqrt{\frac{c}{n}})$ | [6] |

There are a few works studying the behavior of various measures. For instance, [9] analyzed the Bayes consistency of various approaches for HL and RL. [5] provided a unified view of different measures. [10] devoted to study the consistency of reduction approaches for precision@$k$ and recall@$k$. Although they provide valuable insights, the generalization analysis of the algorithms on different measures is still largely open. Furthermore, there is another counter-intuitive observation [11] that in small label space cases, algorithms aiming to optimize HL often have better performance on the SA measure than the algorithms that optimize SA directly. This is inconsistent with the existing theoretical results [12] that SA and HL are conflicting measures — algorithms aiming to optimize HL would perform poorly if evaluated on SA, and vice versa. Although it can provide some insights for existing learning algorithms, the analysis [12] has limitations by assuming that the hypothesis space is unconstrained and the conditional distribution $P(\mathbf{y}|\mathbf{x})$ is known. These assumptions are not held in a realistic algorithm, where a constrained parametric hypothesis space is used and $P(\mathbf{y}|\mathbf{x})$ is unknown.

This paper provides an attempt to fill up this gap by analyzing the generalization bounds for the learning algorithms on various measures, including HL, SA, and RL. These bounds provide insights to explain the aforementioned observations. By avoiding the unwarranted assumptions of previous work, our analysis provides more insights and guidance for learning algorithms in practice. Specifically, here we focus on kernel-based learning algorithms which have been widely-used for MLC [6, 3, 13, 14, 15]. For analysis convenience and fair comparison, we also propose a new algorithm aiming to directly optimize SA with its (convex) surrogate loss function. The main techniques are based on Rademacher complexity [16, 17] and the recent vector-contraction inequality [18]. Note that, different from that for traditional binary or multi-classification problems which are usually analyzed for only one measure (e.g. stand zero-one loss [19]), our generalization analysis for MLC needs to be for both HL and SA. Besides, our analysis can also be extended to analyze other measures, such as RL. To the best of our knowledge, this is the first to provide the generalization error bounds for the learning algorithms between these measures for MLC, including two typical methods Binary Relevance [3] and Rank-SVM [6].

Our main results are summarized in Table 1. We can see that the number of labels (i.e., $c$) plays an important role in the generalization error bounds, which is often ignored in previous work. Besides, we can observe that the algorithm (i.e., Algorithm $\mathcal{A}^h$) aiming for optimize HL has a learning guarantee for HL which is independent with $c$. Furthermore, it also has a learning guarantee for the SA measure which depends on at most $O(c)$. In contrast, when directly optimizing SA with its surrogate loss (i.e., Algorithm $\mathcal{A}^s$), it can have learning guarantees depending on $O(\sqrt{c})$ for both HL and SA. This explains the phenomenon that when the label space is not large, optimizing HL with its surrogate loss can have promising performance for SA. Besides, when the label space is large, optimizing SA directly would enjoy its superiority on SA. Our experimental results also support this theoretical analysis. Interestingly, we also find that optimizing RL with its surrogate loss (i.e. Algorithm $\mathcal{A}^r$) has a learning guarantee on RL which depends on $O(\sqrt{c})$ (See Appendix D for the guarantees of corresponding algorithms for SA and RL).

Overall, our contributions are: (1) We provide the generalization bounds for the corresponding algorithms on various measures, i.e. HL, SA, and RL. Besides, the inequalities between these (actual and surrogate) losses are introduced, which can be used for the learning guarantees between these measures and can also help the analysis extend to other forms of hypothesis classes; (2) based on the theoretical analysis, we explain the phenomenon when in small label space case, optimizing HL with its surrogate loss can have better performance on the SA measure than directly optimizing SA with its surrogate loss; and (3) the experimental results support our theoretical analysis.

The rest of paper is organized as follows. Section 2 introduces the MLC setting and its evaluation measures. Section 3 introduces the main assumptions, theorem, and learning algorithms used in the subsequent analysis. Section 4 presents the learning guarantees of corresponding algorithms for HL and SA. Section 5 presents the learning guarantees of corresponding algorithms for HL and RL. Section 6 reports the experimental results. Section 7 introduces more discussions, and Section 8 concludes this paper.

## 2 Preliminaries

In this section, we introduce the problem setting of MLC and its evaluation measures that we focus on here.

**Notations**. Let the bold-face letters denote for vectors or matrices. For a matrix $\mathbf{A}$, $\mathbf{a}_i$, $\mathbf{a}^j$ and $a_{ij}$ denote its $i$-th row, $j$-th column, and $(i,j)$-th element respectively. For a function $g : \mathbb{R} \to \mathbb{R}$ and a matrix $\mathbf{A} \in \mathbb{R}^{m \times n}$, define $g(\mathbf{A}) : \mathbb{R}^{m \times n} \to \mathbb{R}^{m \times n}$, where $g(\mathbf{A})_{ij} = g(a_{ij})$. $\text{Tr}(\cdot)$ denotes the trace operator for a square matrix. $[\![\pi]\!]$ denotes the indicator function, i.e., it returns 1 when the proposition $\pi$ holds and 0 otherwise. $sgn(x)$ returns 1 when $x > 0$ and $-1$ otherwise. $[n]$ denotes the set $\{1, ..., n\}$.

### 2.1 Problem setting

Given a training set $S = \{(\mathbf{x}_i, \mathbf{y}_i)\}_{i=1}^n$ which is sampled i.i.d. from the distribution $D$ over $\mathcal{X} \times \{-1, +1\}^c$, where $\mathbf{x}_i \in \mathcal{X} \subset \mathbb{R}^d$ is the input, $d$ is the feature dimension, $\mathbf{y}_i \in \{-1, +1\}^c$ is the corresponding label vector, $c$ is the number of potential labels, and $n$ is the number of data points. Besides, $y_{ij} = 1$ (or $-1$) indicates that the $j$-th label is relevant (or irrelevant) with $\mathbf{x}_i$. The goal of MLC is to learn a multi-label classifier $H : \mathbb{R}^d \longrightarrow \{-1, +1\}^c$.

### 2.2 Evaluation measures

To solve the MLC task, one common approach is to first learn a real-valued mapping function $f = [f_1, ..., f_c] : \mathbb{R}^d \longrightarrow \mathbb{R}^c$ and then get the classifier $H(\mathbf{x}) = sgn([\![f(\mathbf{x}) \geq T(\mathbf{x})]\!])$ by use of a thresholding value function $T$. For simplicity, we denote the classifier $H(\mathbf{x}) = t \circ f(\mathbf{x}) = t(f(\mathbf{x}))$, where $t$ is the thresholding function induced by $T$. Besides, note that many algorithms, such as Binary Relevance, just set the thresholding function $T(\mathbf{x}) = 0$ and the classifier becomes $H(\mathbf{x}) = sgn \circ f(\mathbf{x})$.

To evaluate algorithms for MLC, there are many measures. Here we focus on three widely-used measures, i.e., *Hamming Loss*, *Subset Accuracy* and *Ranking Loss*, as defined below[2].

**Hamming Loss** :
$$L_h^{0/1}(t \circ f(\mathbf{x}), \mathbf{y}) = \frac{1}{c} \sum_{j=1}^c [\![t(f_j(\mathbf{x}_i)) \neq \mathbf{y}_j]\!]. \tag{1}$$

For the classifier $H(\mathbf{x}) = sgn \circ f(\mathbf{x})$, its surrogate loss can be defined as:
$$L_h(f(\mathbf{x}), \mathbf{y}) = \frac{1}{c} \sum_{j=1}^c \ell(\mathbf{y}_j f_j(\mathbf{x})), \tag{2}$$

where the base (convex surrogate) loss function $\ell(u)$ can be many popular point-wise loss functions, such as the hinge loss $\ell(u) = \max(0, 1 - u)$ or the logistic loss $\ell(u) = \ln(1 + \exp(-u))$. Besides, we assume the base loss function upper bounds the original 0/1 loss, i.e., $[\![t(f_j(\mathbf{x}_i)) \neq \mathbf{y}_j]\!] \leq \ell(\mathbf{y}_j f_j(\mathbf{x}))$[3].

**Subset Loss** :
$$L_s^{0/1}(t \circ f(\mathbf{x}), \mathbf{y}) = \max_{j \in [c]} \{[\![t(f_j(\mathbf{x}_i)) \neq \mathbf{y}_j]\!]\}. \tag{3}$$

The measure *Subset Accuracy* is equal to $1 - L_s^{0/1}$, where maximizing the *Subset Accuracy* is equivalent to minimize the *Subset Loss*. For the classifier $H(\mathbf{x}) = sgn \circ f(\mathbf{x})$, its (convex) surrogate loss can be defined as:
$$L_s(f(\mathbf{x}), \mathbf{y}) = \max_{j \in [c]} \{\ell(\mathbf{y}_j f_j(\mathbf{x}))\}. \tag{4}$$

**Ranking Loss** : 
$$L_r^{0/1}(f(\mathbf{x}), \mathbf{y}) = \frac{1}{|Y^+||Y^-|} \sum_{p \in Y^+} \sum_{q \in Y^-} [\![f_p(\mathbf{x}) \leq f_q(\mathbf{x})]\!], \tag{5}$$

where $Y^+$ (or $Y^-$) denotes the relevant (or irrelevant) label index set associated with $\mathbf{x}$, and $|\cdot|$ denotes the set cardinality. Besides, its surrogate loss can be defined as:

$$L_r(f(\mathbf{x}), \mathbf{y}) = \frac{1}{|Y^+||Y^-|} \sum_{p \in Y^+} \sum_{q \in Y^-} \ell(f_p(\mathbf{x}) - f_q(\mathbf{x})). \tag{6}$$

There are some relationships between these loss functions. For clarity, we discuss it in the following sections, which is used for the proof of learning guarantees between them.

## 3 Generalization analysis techniques

In this section, we introduce the main assumptions, theorem, and learning algorithms used in the subsequent analysis.

Define a surrogate loss function $L : \mathbb{R}^c \times \{-1, +1\}^c \to \mathbb{R}_+$, and a vector-valued function class $\mathcal{F} = \{f : \mathcal{X} \mapsto \mathbb{R}^c\}$. Then, for a score function $f \in \mathcal{F}$ and its induced classifier $H(\mathbf{x}) = t \circ f(\mathbf{x})$, the true $(0/1)$ expected risk, surrogate expected risk and empirical risk are defined as follows:

$$R_{0/1}(H) = \mathop{\mathbb{E}}_{(\mathbf{x}, \mathbf{y}) \sim D} [L^{0/1}(H(\mathbf{x}), \mathbf{y})] \quad R(f) = \mathop{\mathbb{E}}_{(\mathbf{x}, \mathbf{y}) \sim D} [L(f(\mathbf{x}), \mathbf{y})] \quad \hat{R}_S(f) = \frac{1}{n} \sum_{i=1}^{n} L(f(\mathbf{x}_i), \mathbf{y}).$$

Besides, we use a superscript to distinguish different loss functions. For instance, $\hat{R}_S^h(f)$, $\hat{R}_S^s(f)$, and $\hat{R}_S^r(f)$ denote the empirical Hamming, Subset, and Ranking risk respectively.

In this paper, we focus on kernel-based learning algorithms and utilize Rademacher complexity [16, 17] and the recent vector-contraction inequality [18] to analyze the generalization error bounds for the algorithms. Note that, the local Rademacher complexity [20] can be used to get tighter bounds to improve the learning algorithms, but that is not our main focus. Here we concentrate on the learning guarantees between different measures and analyze them in the same framework for fair comparisons. Due to the space limit, we defer the background about Rademacher complexity and the contraction inequality to Appendix A.1.

We first introduce the common assumptions as follows.

**Assumption 1** (The common assumptions)**.**

*(1) Let $\kappa : \mathcal{X} \times \mathcal{X} \to \mathbb{R}$ be a Positive Definite Symmetric (PSD) kernel and $\phi : \mathbf{x} \to \mathbb{H}$ be a feature mapping associated with $\kappa$, where $\mathbb{H}$ is its associated reproducing kernel Hilbert space (RKHS). Here, we consider the following kernel-based hypothesis set:*

$$\mathcal{F} = \{\mathbf{x} \longmapsto \mathbf{W}^\top \phi(\mathbf{x}) : \mathbf{W} = (\mathbf{w}_1, \dots, \mathbf{w}_c)^\top, \|\mathbf{W}\|_{\mathbb{H},2} \leq \Lambda\}, \tag{7}$$

*where $\|\mathbf{W}\|_{\mathbb{H},2} = (\sum_{j=1}^{c} \|\mathbf{w}_j\|_{\mathbb{H}}^2)^{1/2}$. For notational clarity, we denote $\|\mathbf{W}\|_{\mathbb{H},2}$ by $\|\mathbf{W}\|$ in the following.*

*(2) The training dataset $S$ is an i.i.d. sample of size $n$ drawn from the distribution $D$, where $\exists\, r > 0$, it satisfies $\kappa(\mathbf{x}, \mathbf{x}) \leq r^2$ for all $\mathbf{x} \in \mathcal{X}$.*

*(3) The base loss function $\ell(u)$ is $\rho$-Lipschitz continuous and bounded by $B$.*

Note that, although our subsequent analysis is based on the kernel-based hypothesis set, it can also be extended to other forms of hypothesis set, such as neural networks [16, 21]. Besides, the linear hypothesis can be viewed as a special case of the kernel-based one where the kernel function is linear. Furthermore, for the base loss function $\ell(u)$, the assumption can be satisfied for many popular point-wise loss functions. For instance, the widely-used hinge loss $\ell(u) = \max(0, 1 - u)$ and the logistic loss $\ell(u) = \ln(1 + \exp(-u))$ are both $1-$Lipschitz.

Besides, we give an extra assumption for the following discussion about ranking-based learning algorithms.

**Assumption 2** (For the ranking-based algorithms). *For the ranking-based classifier $H(\mathbf{x}) = t \circ f(\mathbf{x})$, assume the thresholding function $t(f)$ splits the label list into two parts based on the score function in the non-ascending order. Besides, let the oracle optimal thresholding function be $t^*(f)$, which gets the best Hamming Loss for a given score function.*

Next, we analyze the Lipschitz constant and upper bound of the surrogate loss function in the following lemma, which is used for the subsequent analysis.

**Lemma 1** (The property of the surrogate loss function; full proof in Appendix A.2.). *Assume that the base loss function $\ell(u)$ is $\rho$-Lipschitz continuous and bounded by $B$. Then, the surrogate Hamming Loss (2) is $\frac{\rho}{\sqrt{c}}$-Lipschitz, the surrogate Ranking Loss (6) is $\rho$-Lipschitz, and the surrogate Subset Loss (4) is $\rho$-Lipschitz w.r.t. the first argument. Besides, they are all bounded by $B$.*

Furthermore, we give the base theorem used in the subsequent generalization analysis, as follows.

**Theorem 1** (The base theorem for generalization analysis; full proof in Appendix A.3). *Assume the loss function $L : \mathbb{R}^c \times \{-1, +1\}^c \to \mathbb{R}_+$ is $\mu$-Lipschitz continuous w.r.t. the first argument and bounded by $M$. Besides, (1) and (2) in* Assumption 1 *are satisfied. Then, for any $\delta > 0$, with probability at least $1 - \delta$ over the draw of an i.i.d. sample $S$ of size $n$, the following generalization bound holds for all $f \in \mathcal{F}$:*

$$R(f) \leq \hat{R}_S(f) + 2\sqrt{2}\mu\sqrt{\frac{c\Lambda^2 r^2}{n}} + 3M\sqrt{\frac{\log \frac{2}{\delta}}{2n}}. \tag{8}$$

Last, for the clarity of subsequent discussions, we introduce the learning algorithms which directly optimize the measures of Hamming Loss, Subset Loss, and Ranking Loss with their corresponding surrogate loss functions, denoted by $\mathcal{A}^h$, $\mathcal{A}^s$, and $\mathcal{A}^r$ respectively as follows:

$$\mathcal{A}^h : \ \min_{\mathbf{W}} \ \frac{1}{n} \sum_{i=1}^{n} L_h(f(\mathbf{x}_i), \mathbf{y}_i) + \lambda \|\mathbf{W}\|^2, \tag{9}$$

$$\mathcal{A}^s : \ \min_{\mathbf{W}} \ \frac{1}{n} \sum_{i=1}^{n} L_s(f(\mathbf{x}_i), \mathbf{y}_i) + \lambda \|\mathbf{W}\|^2, \tag{10}$$

$$\mathcal{A}^r : \ \min_{\mathbf{W}} \ \frac{1}{n} \sum_{i=1}^{n} L_r(f(\mathbf{x}_i), \mathbf{y}_i) + \lambda \|\mathbf{W}\|^2. \tag{11}$$

## 4  Learning guarantees between Hamming and Subset Loss

In this section, we first analyze the relationships between Hamming and Subset Loss. Then, we analyze the leaning guarantees of algorithm $\mathcal{A}^h$ w.r.t. the measures of Hamming and Subset Loss. Last, we analyze the learning guarantees of algorithm $\mathcal{A}^s$ w.r.t. these two measures.

First, we analyze the relationship between them, which is shown as follows.

**Lemma 2** (The relationship between Hamming and Subset Loss). *For the classifier $H(\mathbf{x}) = sgn \circ f(\mathbf{x})$, the following inequalities hold:*

$$L_h^{0/1}(H(\mathbf{x}), \mathbf{y}) \leq L_s^{0/1}(H(\mathbf{x}), \mathbf{y}) \leq L_s(f(\mathbf{x}), \mathbf{y}), \tag{12}$$

$$L_s^{0/1}(H(\mathbf{x}), \mathbf{y}) \leq cL_h^{0/1}(H(\mathbf{x}), \mathbf{y}) \leq cL_h(f(\mathbf{x}), \mathbf{y}). \tag{13}$$

The proof is similar to [12]. For completeness, we add it to Appendix B.1. From this lemma, we can observe that when optimizing Subset Loss with its surrogate loss, it actually also optimizes an upper bound of Hamming Loss. Besides, when optimizing Hamming Loss with its surrogate loss, it also optimizes an upper bound for Subset Loss which depends on $O(c)$. This can be used to provide the learning guarantees between them.

## 4.1 Learning guarantees of Algorithm $\mathcal{A}^h$

The learning guarantee of $\mathcal{A}^h$ w.r.t. Hamming Loss is shown in the following theorem.

**Theorem 2** (Optimize Hamming Loss, Hamming Loss bound). *Assume the loss function $L = L_h$, where $L_h$ is defined in Eq.(2). Besides,* Assumption 1 *is satisfied. Then, for any $\delta > 0$, with probability at least $1 - \delta$ over S, the following generalization bound in terms of* Hamming Loss *holds for all $f \in \mathcal{F}$:*

$$R_{0/1}^h(sgn \circ f) \leq \hat{R}_S^h(f) + 2\sqrt{2}\rho\sqrt{\frac{\Lambda^2 r^2}{n}} + 3B\sqrt{\frac{\log \frac{2}{\delta}}{2n}}. \tag{14}$$

*Proof.* (sketch; full proof in Appendix B.2) The key step is to apply Theorem 1 and Lemma 1. Besides, the inequality $R_{0/1}^h(sgn \circ f) \leq R^h(f)$ holds. □

Then, we can get the generalization bound for the classical Binary Relevance [3] on Hamming Loss, which we defer it in Appendix B.3. From the above theorem, we can observe that $\mathcal{A}^h$ has a good learning guarantee for Hamming Loss independent of $c$. Besides, [9] has shown it is Bayes consistent for Hamming Loss, which confirms its superiority for Hamming Loss. Moreover, $\mathcal{A}^h$ also has a learning guarantee for Subset Loss by the following theorem.

**Theorem 3** (Optimize Hamming Loss, Subset Loss bound). *Assume the loss function $L = cL_h$, where $L_h$ is defined in Eq.(2). Besides,* Assumption 1 *is satisfied. Then, for any $\delta > 0$, with probability at least $1 - \delta$ over S, the following generalization bound in terms of* Subset Loss *holds for all $f \in \mathcal{F}$:*

$$R_{0/1}^s(sgn \circ f) \leq c R_{0/1}^h(sgn \circ f) \leq c\hat{R}_S^h(f) + 2\sqrt{2}\rho c\sqrt{\frac{\Lambda^2 r^2}{n}} + 3Bc\sqrt{\frac{\log \frac{2}{\delta}}{2n}}. \tag{15}$$

*Proof.* (sketch; full proof in Appendix B.4) The main idea is to apply the Theorem 1, Lemma 1 and 2. Besides, $R_{0/1}^h(sgn \circ f) \leq R^h(f)$. □

From the above theorem, we can observe that $\mathcal{A}^h$ has a bound on Subset Loss which depends on $O(c)$. When $c$ is small, its performance for Subset Loss would probably enjoy its good leaning guarantee for Hamming Loss.

## 4.2 Learning guarantees of Algorithm $\mathcal{A}^s$

The following theorem provides the learning guarantees of $\mathcal{A}^s$ w.r.t. the measures of Subset and Hamming Loss. The proof is similar to those for Theorem 2 & 3 in Section 4.1.

**Theorem 4** (Optimize Subset Loss, Subset and Hamming Loss bounds). *Assume the loss function $L = L_s$, where $L_s$ is defined by (4). Besides,* Assumption 1 *is satisfied. Then, for any $\delta > 0$, with probability at least $1 - \delta$ over S, the following generalization bounds in terms of* Subset and Hamming Loss *hold for all $f \in \mathcal{F}$:*

$$R_{0/1}^h(sgn \circ f) \leq R_{0/1}^s(sgn \circ f) \leq \hat{R}_S^s(f) + 2\sqrt{2}\rho\sqrt{\frac{c\Lambda^2 r^2}{n}} + 3B\sqrt{\frac{\log \frac{2}{\delta}}{2n}}. \tag{16}$$

The full proof is in Appendix B.5. From this theorem, we can observe $\mathcal{A}^s$ has the same bounds for Subset and Hamming Loss both depending on $O(\sqrt{c})$. Intuitively, the learning guarantee of $\mathcal{A}^s$ for Hamming Loss comes from its learning guarantee for Subset Loss.

## 4.3 Comparisons

For the same hypothesis set, $\hat{R}_S^s(f)$ is usually harder to train than $\hat{R}_S^h(f)$, which makes $\hat{R}_S^h(f)$ smaller[4]. For Hamming Loss, comparing the bounds for $\mathcal{A}^h$ (i.e. InEq.(14) in Theorem 2) and $\mathcal{A}^s$ (i.e. InEq.(16) in Theorem 4), we can conclude that $\mathcal{A}^h$ has tighter bound than $\mathcal{A}^s$, thus $\mathcal{A}^h$ would perform better than $\mathcal{A}^s$. For Subset Loss, comparing the bounds[5] for $\mathcal{A}^h$ (i.e. InEq.(15) in Theorem

3) and $\mathcal{A}^s$ (i.e. InEq.(16) in Theorem 4), we can conclude that, in the large label space case, $\mathcal{A}^s$ would probably perform better than $\mathcal{A}^h$; however, in the small label space case, $\mathcal{A}^h$ can enjoy its good learning guarantee for Hamming Loss while $\mathcal{A}^s$ cannot, thus $\mathcal{A}^h$ would probably have better performance than $\mathcal{A}^s$. Experimental results also support our theoretical analysis.

## 5  Learning guarantees between Hamming and Ranking Loss

In this section, we first analyze the relationships between Hamming and Ranking Loss. Then, we analyze the learning guarantee of $\mathcal{A}^h$ on the Ranking Loss measure. Last, we analyze the learning guarantee of $\mathcal{A}^r$ on these two measures.

First, we analyze the relationship between Hamming and Ranking Loss, which is shown as follows.

**Lemma 3** (The relationship between Hamming and Ranking Loss)**.** *For the Hamming and Ranking Loss, the following inequality holds:*

$$L_r^{0/1}(f(\mathbf{x}), \mathbf{y}) \leq cL_h^{0/1}(sgn \circ f(\mathbf{x}), \mathbf{y}) \leq cL_h(f(\mathbf{x}), \mathbf{y}). \tag{17}$$

*Further, if* Assumption 2 *is satisfied, the following inequality holds:*

$$L_h^{0/1}(t^* \circ f(\mathbf{x}), \mathbf{y}) \leq cL_r^{0/1}(f(\mathbf{x}), \mathbf{y}) \leq cL_r(f(\mathbf{x}), \mathbf{y}). \tag{18}$$

The full proof is in Appendix C.1[6]. From this lemma, we can observe that Ranking Loss is upper bounded by Hamming Loss times the label size[7]. Similarly, Hamming Loss is upper bounded by Ranking Loss times the label size. Thus, when optimizing one measure with its surrogate, it also optimizes an upper bound for another measure and provides its learning guarantee.

### 5.1  Learning guarantee of Algorithm $\mathcal{A}^h$

The learning algorithm $\mathcal{A}^h$ has a learning guarantee w.r.t. Ranking Loss, as shown by the following theorem.

**Theorem 5** (Optimize Hamming Loss, Ranking Loss bound)**.** *Assume the loss function $L = cL_h$, where $L_h$ is defined in Eq.(2). Besides,* Assumption 1 *is satisfied. Then, for any $\delta > 0$, with probability at least $1 - \delta$ over S, the following generalization bound in terms of* Ranking Loss *holds for all $f \in \mathcal{F}$:*

$$R_{0/1}^r(f) \leq cR_{0/1}^h(sgn \circ f) \leq c\hat{R}_S^h(f) + 2\sqrt{2}\rho c\sqrt{\frac{\Lambda^2 r^2}{n}} + 3Bc\sqrt{\frac{\log \frac{2}{\delta}}{2n}}. \tag{19}$$

The full proof is in Appendix C.2. From this theorem, we can observe that $\mathcal{A}^h$ has a learning guarantee for Ranking Loss depending on $O(c)$. When $c$ is small, $\mathcal{A}^h$ can have promising performance for Ranking Loss.

### 5.2  Learning guarantee of Algorithm $\mathcal{A}^r$

The learning algorithm $\mathcal{A}^r$ has a learning guarantee w.r.t. Ranking Loss as follows.

**Theorem 6** (Optimize Ranking Loss, Ranking Loss bound)**.** *Assume the loss function $L = L_r$, where $L_r$ is defined in Eq.(6). Besides,* Assumption 1 *is satisfied. Then, for any $\delta > 0$, with probability at least $1 - \delta$ over S, the following generalization bound in terms of* Ranking Loss *holds for all $f \in \mathcal{F}$:*

$$R_{0/1}^r(f) \leq \hat{R}_S^r(f) + 2\sqrt{2}\rho\sqrt{\frac{c\Lambda^2 r^2}{n}} + 3B\sqrt{\frac{\log \frac{2}{\delta}}{2n}}. \tag{20}$$

The full proof is in Appendix C.3. Then, we can get the learning guarantee of the classical Rank-SVM [6] (See Appendix C.4). To the best of our knowledge, this is the first to provide its generalization bound on Ranking Loss. From the above theorem, we can observe that $\mathcal{A}^r$ has a learning guarantee for Ranking Loss depending on $O(\sqrt{c})$, which illustrates its superiority for large label space compared with $\mathcal{A}^h$. Besides, similar to the analysis between $\mathcal{A}^h$ and $\mathcal{A}^r$, we can also conclude that $\mathcal{A}^r$ performs better than $\mathcal{A}^s$ w.r.t. Ranking Loss. $\mathcal{A}^r$ also has a learning guarantee w.r.t. Hamming Loss (See Appendix C.5).

## 6   Experiments

The purpose of this paper is to provide a generalization analysis of learning algorithms for different measures and take insights to explain the aforementioned observations. Thus, for the experiments, the goal is to validate our theoretical results rather than illustrating the performance superiority of our proposed algorithm. Therefore, we focus on two algorithms, i.e. optimizing Hamming Loss ($\mathcal{A}^h$) and optimizing Subset Loss ($\mathcal{A}^s$), and evaluate them in terms of Subset Accuracy[1] and Hamming Loss on datasets with different label sizes.

Specifically, six commonly used benchmark datasets from various domains and different label sizes are used: image (image, 5), emotions (music, 6), scene (image, 6), enron (text, 53), rcv1-subset1 (text, 101), and bibtex (text, 159), which are downloaded from the open websites[2]. Besides, for the first three datasets, we normalize the input to mean $= 0$ and deviation $= 1$. For $\mathcal{A}^h$ and $\mathcal{A}^s$, we take the linear models with the hinge base loss function for simplicity, and utilize SVRG-BB [22] to efficiently train the models[3]. We conduct 3-fold cross-validation on each dataset, where the hyper-parameter $\lambda$ is searched in $\{10^{-4}, 10^{-3}, \cdots, 10\}$.

Table 2 reports the results in terms of Hamming Loss. We can observe $\mathcal{A}^h$ performs better than $\mathcal{A}^s$. This validates our theoretical analysis that $\mathcal{A}^h$ has tighter generalization bound than $\mathcal{A}^s$ on Hamming Loss.

Table 2: The results of various datasets in terms of Hamming Loss (mean $\pm$ std). The smaller the value, the better. Best results are in bold. The numbers in brackets represent the label size.

| Dataset | emotions(6) | image(5) | scene(6) | enron(53) | rcv1-subset1(101) | bibtex(159) |
|---|---|---|---|---|---|---|
| $\mathcal{A}^h$ | $\mathbf{0.202 \pm 0.019}$ | $\mathbf{0.180 \pm 0.002}$ | $\mathbf{0.103 \pm 0.011}$ | $\mathbf{0.047 \pm 0.001}$ | $\mathbf{0.027 \pm 0.000}$ | $\mathbf{0.013 \pm 0.000}$ |
| $\mathcal{A}^s$ | $0.224 \pm 0.015$ | $0.214 \pm 0.013$ | $0.142 \pm 0.009$ | $0.055 \pm 0.001$ | $0.032 \pm 0.000$ | $0.015 \pm 0.000$ |

Besides, Table 3 reports the results in terms of Subset Accuracy. We can observe that for small label space datasets, $\mathcal{A}^h$ performs better than $\mathcal{A}^s$. In contrast, for relatively large label space datasets, $\mathcal{A}^s$ performs better than $\mathcal{A}^h$. This also validates our theoretical analysis results.

Table 3: The results of various datasets in terms of Subset Accuracy (mean $\pm$ std). The larger the value, the better. Best results are in bold.

| Dataset | emotions(6) | image(5) | scene(6) | enron(53) | rcv1-subset1(101) | bibtex(159) |
|---|---|---|---|---|---|---|
| $\mathcal{A}^h$ | $\mathbf{0.288 \pm 0.026}$ | $\mathbf{0.471 \pm 0.004}$ | $\mathbf{0.628 \pm 0.035}$ | $\mathbf{0.143 \pm 0.005}$ | $0.079 \pm 0.008$ | $0.190 \pm 0.001$ |
| $\mathcal{A}^s$ | $0.240 \pm 0.021$ | $0.396 \pm 0.031$ | $0.515 \pm 0.032$ | $0.133 \pm 0.013$ | $\mathbf{0.111 \pm 0.004}$ | $\mathbf{0.198 \pm 0.001}$ |

## 7   Discussions

There are two competing approach frameworks [23, 24] w.r.t. a loss $L^{0/1}$ for MLC: 1) the decision-theoretic approach (DTA) fits a probabilistic model to estimate $P(\mathbf{y}|\mathbf{x})$ during training, followed by an inference phase for every test instance via the optimal strategy w.r.t. $L^{0/1}$; 2) the empirical utility maximization (EUM) approach optimizes $L^{0/1}$ with its surrogate loss to find a classifier in a constrained parametric hypothesis space during training. The analysis in [12] is mainly under the DTA framework, while ours is under the EUM framework and complementary to [12]. Below, we discuss the pros and cons of each one in detail.

**Pros and cons of the analysis in [12]:** [12] can provide much insight for the DTA framework although there is still a gap between the actual $P(\mathbf{y}|\mathbf{x})$ and its estimated one through many parametric methods (e.g., probabilistic classifier chains [25]). In contrast, it may offer little insight for the EUM framework (e.g., Binary Relevance which directly optimizes HL with its surrogate loss). Specifically, [12] assumes that the hypothesis space is unconstrained to allow $P(\mathbf{y}|\mathbf{x})$ known, and gets the Bayes-optimal classifiers w.r.t. HL (i.e. $\mathbf{h}_H^*$) and SA (i.e. $\mathbf{h}_s^*$) by their corresponding optimal strategy. Then, it analyzes the regret (a.k.a excess risk) upper bounds of $\mathbf{h}_H^*$ and $\mathbf{h}_s^*$ in terms of SA (i.e., Proposition 4) and HL (i.e., Proposition 5) respectively, and finds the bounds are large, which concludes that HL and SA conflict with each other.

**Pros and cons of our analysis:** Our analysis can provide much insight for the EUM framework, while it may offer little insight for the DTA framework. Specifically, we directly analyze the generalization bounds for the learning algorithms w.r.t. many measures. Although here we consider the kernel-based hypothesis class, which includes the linear and non-linear model by specifying different kernel functions, our analysis can be extended to other forms of hypothesis classes. Meanwhile, our analysis misses the aspect of consistency which is a central point of [12]. Besides, our analysis is for a specific model that may be constrained for optimizing the SA measure.

There are many methods that aim to optimize the SA measure. Since the Bayes decision for SA is based on the joint mode of the conditional distribution instead of the marginal modes as for HL [12], the methods optimizing SA need to model conditional label dependencies to estimate (at least implicitly) the joint distribution of labels. One typical method is the structured SVM [26, 27][1], which enables incorporating label dependencies to the joint feature space defined on $\mathbf{y}$ and $\mathbf{x}$. The struct hinge loss for each sample $(\mathbf{x}_i, \mathbf{y}_i)$ is $\max_{\mathbf{y} \in \{-1, +1\}^c} \{0, L_s^{0/1}(\mathbf{y}, \mathbf{y}_i) + \langle \Psi(\mathbf{x}, \mathbf{y}), \theta \rangle - \langle \Psi(\mathbf{x}, \mathbf{y}_i), \theta \rangle \}$, which is defined over all $2^c$ label vectors and a convex upper bound for $L_s^{0/1}(\mathbf{y}, \mathbf{y}_i)$. In comparison, our proposed (convex) surrogate hinge loss $L_s$ for SA is $\max_{j \in [c]} \{\max\{0, 1 - y_{ij} \langle \phi(\mathbf{x}_i), \mathbf{w}^j \rangle\}\}$, which is defined over the label size $c$ and thus has better computational efficiency than the struct hinge loss. Although $L_s$ also incorporates label dependencies, it is interesting to test whether $L_s$ is sufficient to find the joint mode of the distribution. We will study its performance by comparing with other state-of-the-art methods for SA including the structured SVM in the future.

Besides, Label Powerset (LP) is another representative method for optimizing SA. It transforms MLC into a multi-label classification problem where each subset can be viewed as a new class and eventually constructs an exponential number of classes (i.e. $2^c$) in total. Based on the theoretical results [18, 28] for multi-class classification, LP has a generalization error bound w.r.t. SA which depends on $O(\sqrt{2^c})$.[2] This explains that it would perform poorly when the label size is large, and inspires Random k-Labelsets (RAKEL) [7] to boost the performance by combining ensemble techniques and LP with a small label space. Finally, some work [29] claims that algorithms designed for SA perform well for HL, which needs more exploration to explain it.

# 8 Conclusions

This paper attempts to theoretically analyze the effects of learning algorithms on the measures of Hamming, Subset, and Ranking Loss by providing the generalization bounds for algorithms on these measures. Through the analysis, we find that the label size has an important effect on the learning guarantees of an algorithm for different measures. Besides, we take insights from the theoretical results to explain the phenomenon that in small label space case, optimizing Hamming Loss with its surrogate loss can perform well for Subset Loss. Experimental results also support our theory findings. In the future, our analysis techniques can be extended for the generalization analysis of other measures. Besides, how to make these bounds tighter will inspire more effective learning algorithms.

## Broader Impact

As a theoretical research, this work will potentially provide insights for developing better algorithms for multi-label classification, while without explicit negative consequences to our society.

## Acknowledgments and Disclosure of Funding

We thank Chongxuan Li for valuable discussions. We also thank all four reviewers for their insightful comments and meta-reviewer for the extensive invaluable comments to improve the paper quality. This work was supported by the National Key Research and Development Program of China (No.2017YFA0700904), NSFC Projects (Nos. 61620106010, U19B2034, U1811461), Beijing Academy of Artificial Intelligence (BAAI), Tsinghua-Huawei Joint Research Program, a grant from Tsinghua Institute for Guo-Qiang, Tiangong Institute for Intelligent Computing, and the NVIDIA NVAIL Program with GPU/DGX Acceleration.

## Footnotes

[1]Subset Loss is equal to $1-$ Subset Accuracy

[2]Our definition is over a sample and can be averaged over many samples.

[3]The original logistic loss can be simply changed to $\ell(u) = \log_2(1 + \exp(-u))$ to satisfy this condition.

[4]Although we cannot formally express this, experimental results support it.

[5]Note that, in practice, it probably makes no sense to directly compare the absolute values for the bounds of $\mathcal{A}^h$ and $\mathcal{A}^s$ on Subset Loss because the first items of bounds would probably be close to or bigger than 1. However, we can still take insights from them to get the dependent variables.

[6]Note that, the proof is nontrivial, especially for the second inequality.

[7]Precisely, it also depends on the ratio of relevant and irrelevant labels (See Appendix C.1). Note that the following analyses are based on the label size and we believe they can be improved by involving the ratio of relevant and irrelevant labels.

[1]For usual practice, we don't utilize Subset Loss although they are equivalent.

[2]http://mulan.sourceforge.net/datasets-mlc.html and http://palm.seu.edu.cn/zhangml/

[3]Note that the models are both convex optimization problems.

[1]Note that, although F-score is optimized in [27], we can easily adapt it to optimize SA by replacing the $\triangle(y, y^n)$ with subset zero-one loss.

[2]Note that, this bound is provided for fair comparison by using the same techniques [18] in this paper and it can be improved to be dependent on $O(c^{\frac{3}{2}})$ by the techniques in [28].

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
