[Supplementary Material]

# Supplementary Material for "Multi-label classification: do Hamming loss and subset accuracy really conflict with each other?"

## A  Generalization analysis techniques

In this section, we review the background on Rademacher complexity [1, 2] and the contraction inequality [3]. Then, we present the bound for the Rademacher complexity of the kernel-based hypothesis set. Last, we provide the detailed proofs for Lemma 1 and Theorem 1.

### A.1  Background on Rademacher complexity and the contraction inequality

**Definition 1** (The loss function space). *For the loss function $L : \mathbb{R}^c \times \{-1, +1\}^c \to \mathbb{R}_+$, the loss function space associated with $\mathcal{F}$ is a family of functions mapping from $(\mathbf{x}, \mathbf{y})$ to $\mathbb{R}_+$, which is as follows:*

$$\mathcal{G} = \{g : (\mathbf{x}, \mathbf{y}) \mapsto L(f(\mathbf{x}), \mathbf{y}) : f \in \mathcal{F}\}.$$

**Definition 2** (The Rademacher complexity of the loss space). *The empirical Rademacher complexity of the loss function space is defined as follows:*

$$\hat{\mathfrak{R}}_S(\mathcal{G}) = \mathbb{E}_{\boldsymbol{\epsilon}}\left[\sup_{g \in \mathcal{G}} \frac{1}{n} \sum_{i=1}^{n} \epsilon_i g(\mathbf{z}_i)\right],$$

*where $\mathbf{z}_i = (\mathbf{x}_i, \mathbf{y}_i)$, and $\boldsymbol{\epsilon} = [\epsilon_1, ..., \epsilon_n]$ in which $\epsilon_i$ is i.i.d. sampled from the Rademacher distribution $Unif(\{-1, +1\})$. Besides, its deterministic counterpart is $\mathfrak{R}_n(\mathcal{G}) = \mathbb{E}_{S \sim D^n}[\hat{\mathfrak{R}}_S(\mathcal{G})]$.*

**Definition 3** (The Rademacher complexity of the hypothesis space). *The empirical Rademacher complexity of the hypothesis space is defined as follows:*

$$\hat{\mathfrak{R}}_S(\mathcal{F}) = \mathbb{E}_{\boldsymbol{\epsilon}}\left[\sup_{f \in \mathcal{F}} \frac{1}{n} \sum_{i=1}^{n} \sum_{j=1}^{c} \epsilon_{ij} f_j(\mathbf{x}_i)\right],$$

*where $\boldsymbol{\epsilon} = [(\epsilon_{ij})] \in \{-1, +1\}^{n \times c}$ in which each element $\epsilon_{ij}$ is i.i.d. sampled from the Rademacher distribution $Unif(\{+1, -1\})$ and $f(\mathbf{x}_i) = [f_1(\mathbf{x}_i), ..., f_c(\mathbf{x}_i)]$. Besides, its deterministic counterpart is $\mathfrak{R}_n(\mathcal{F}) = \mathbb{E}_{S \sim D^n}[\hat{\mathfrak{R}}_S(\mathcal{F})]$.*

**Theorem A.1** ([2]). *Assume $\mathcal{G}$ be a family of functions from $\mathbf{z}$ to $[0, M]$. Then, for any $\delta > 0$, with probability at least $1 - \delta$ over the draw of an i.i.d. sample $S$ of size $n$, the following generalization bound holds for all $g \in \mathcal{G}$:*

$$\mathbb{E}[g(z)] \leq \frac{1}{n} \sum_{i=1}^{n} g(\mathbf{z}_i) + 2\hat{\mathfrak{R}}_S(\mathcal{G}) + 3M\sqrt{\frac{\log \frac{2}{\delta}}{2n}}. \tag{1}$$

**Lemma A.1** (The contraction lemma [3]). *Assume the loss function $L$ is $\mu$-lipschitz w.r.t. the first argument, i.e. $\forall f^1, f^2 \in \mathcal{F}, |L(f^1(\mathbf{x}), \mathbf{y}) - L(f^2(\mathbf{x}), \mathbf{y})| \leq \mu \|f^1(\mathbf{x}) - f^2(\mathbf{x})\|$ always holds. Then, the following inequality holds:*

$$\hat{\mathfrak{R}}_S(\mathcal{G}) \leq \sqrt{2}\mu\hat{\mathfrak{R}}_S(\mathcal{F}). \tag{2}$$

**Theorem A.2.** *Assume the loss function $L : \mathbb{R}^c \times \{-1, +1\}^c \to \mathbb{R}_+$ is $\mu$-Lipschitz continuous w.r.t. the first argument and bounded by $M$. Then, for any $\delta > 0$, with probability at least $1 - \delta$ over the draw of an i.i.d. sample $S$ of size $n$, the following generalization bound holds for all $f \in \mathcal{F}$:*

$$R(f) \le \hat{R}_S(f) + 2\sqrt{2}\mu\hat{\mathfrak{R}}_S(\mathcal{F}) + 3M\sqrt{\frac{\log\frac{2}{\delta}}{2n}}. \tag{3}$$

*Proof.* It is straightforward to get this theorem by applying Theorem A.1 and Lemma A.1. $\qquad\square$

**Lemma A.2** (The Rademacher complexity of the kernel-based hypothesis set)**.** *Assume that there exists $r > 0$ such that $\kappa(\mathbf{x}, \mathbf{x}) \le r^2$ for all $\mathbf{x} \in \mathcal{X}$. Then, for the kernel-based hypothesis set $\mathcal{F} = \{\mathbf{x} \longmapsto \mathbf{W}^\top \phi(\mathbf{x}) : \mathbf{W} = (\mathbf{w}_1, \dots, \mathbf{w}_c)^\top, \|\mathbf{W}\| \le \Lambda\}$, $\hat{\mathfrak{R}}_S(\mathcal{F})$ can be bounded bellow:*

$$\hat{\mathfrak{R}}_S(\mathcal{F}) \le \sqrt{\frac{c\Lambda^2 r^2}{n}}. \tag{4}$$

*Proof.* For the kernel-based hypothesis set $\mathcal{F} = \{\mathbf{x} \longmapsto \mathbf{W}^\top \phi(\mathbf{x}) : \mathbf{W} = (\mathbf{w}_1, \dots, \mathbf{w}_c)^\top, \|\mathbf{W}\| \le \Lambda\}$, the following inequalities about $\hat{\mathfrak{R}}_S(\mathcal{F})$ hold:

$$
\begin{aligned}
\hat{\mathfrak{R}}_S(\mathcal{F}) &= \frac{1}{n}\mathbb{E}_\epsilon\left[\sup_{\|\mathbf{W}\|\le\Lambda}\sum_{i=1}^n\sum_{j=1}^c \epsilon_{ij}\langle\mathbf{w}_j, \phi(\mathbf{x}_i)\rangle\right] \\
&= \frac{1}{n}\mathbb{E}_\epsilon\left[\sup_{\|\mathbf{W}\|\le\Lambda}\sum_{j=1}^c\left\langle\mathbf{w}_j, \sum_{i=1}^n \epsilon_{ij}\phi(\mathbf{x}_i)\right\rangle\right] \\
&= \frac{1}{n}\mathbb{E}_\epsilon\left[\sup_{\|\mathbf{W}\|\le\Lambda}\langle\mathbf{W}, \mathbf{X}_\epsilon\rangle\right] \qquad (\mathbf{X}_\epsilon = [\sum_{i=1}^n\epsilon_{i1}\phi(\mathbf{x}_i), \dots, \sum_{i=1}^n\epsilon_{ic}\phi(\mathbf{x}_i)]) \\
&\le \frac{1}{n}\mathbb{E}_\epsilon\left[\sup_{\|\mathbf{W}\|\le\Lambda}\|\mathbf{W}\|\,\|\mathbf{X}_\epsilon\|\right] \qquad (\textit{Cauchy-Schwarz Inequality}) \\
&= \frac{\Lambda}{n}\mathbb{E}_\epsilon\left[\sum_{j=1}^c\|\sum_{i=1}^n\epsilon_{ij}\phi(\mathbf{x}_i)\|^2\right]^{1/2} \\
&= \frac{\Lambda}{n}\mathbb{E}_\epsilon\left[\sum_{j=1}^c\sum_{p=1}^n\sum_{q=1}^n\epsilon_{pj}\epsilon_{qj}\langle\phi(\mathbf{x}_p), \phi(\mathbf{x}_q)\rangle\right]^{1/2} \\
&= \frac{\Lambda}{n}\mathbb{E}_\epsilon\left[\sum_{j=1}^c\sum_{i=1}^n\langle\phi(\mathbf{x}_i), \phi(\mathbf{x}_i)\rangle\right]^{1/2} \qquad (\forall p \ne q, \mathbb{E}[\epsilon_{pj}\epsilon_{qj}] = \mathbb{E}[\epsilon_{pj}]\mathbb{E}[\epsilon_{qj}] = 0 \textit{ and } \mathbb{E}[\epsilon_{ij}\epsilon_{ij}] = 1) \\
&= \frac{\Lambda\sqrt{c\,\mathrm{Tr}(\mathbf{K})}}{n} \qquad (\kappa(\mathbf{x}_i, \mathbf{x}_i) = \langle\phi(\mathbf{x}_i), \phi(\mathbf{x}_i)\rangle, \mathbf{K} = [\kappa(\mathbf{x}_i, \mathbf{x}_j)] \textit{ is the kernel matrix}) \\
&\le \frac{\sqrt{c\Lambda^2 r^2}}{n}.
\end{aligned}
\tag{5}
$$

$\square$

## A.2 The property of the surrogate loss function

**Lemma 1** (The property of the surrogate loss function)**.** *Assume that the base loss function $\ell(u)$ is $\rho$-Lipschitz continuous and bounded by $B$. Then, the surrogate Hamming Loss Eq.(2) is $\frac{\rho}{\sqrt{c}}$-Lipschitz, the surrogate Ranking Loss Eq.(6) is $\rho$-Lipschitz, and the surrogate Subset Loss Eq.(4) is $\rho$-Lipschitz. Besides, they are all bounded by $B$.*

*Proof.* For notation clarity, we denote $f(\mathbf{x})$ by $f$ in the following. For the surrogate Hamming Loss Eq.(2), $\forall f^1, f^2 \in \mathcal{F}$, the following holds:

$$
\begin{aligned}
&|L_h(f^1, \mathbf{y}) - L_h(f^2, \mathbf{y})| \\
&= \frac{1}{c} \sum_{j=1}^{c} |\ell(f_j^1, \mathbf{y}_j) - \ell(f_j^2, \mathbf{y}_j)| \\
&= \frac{1}{c} \sum_{j=1}^{c} |\ell(\mathbf{y}_j f_j^1) - \ell(\mathbf{y}_j f_j^2)| \\
&\leq \frac{1}{c} \sum_{j=1}^{c} \rho |\mathbf{y}_j f_j^1 - \mathbf{y}_j f_j^2| \qquad\qquad (\ell(u)\ is\ \rho - Lipschitz) \\
&\leq \rho \left[ \frac{1}{c} \sum_{j=1}^{c} |f_j^1 - f_j^2|^2 \right]^{1/2} \qquad\qquad (Jense's\ Inequality) \\
&= \frac{\rho}{\sqrt{c}} \|f^1 - f^2\|.
\end{aligned}
\tag{6}
$$

For the surrogate Ranking Loss Eq.(6), $\forall f^1, f^2 \in \mathcal{F}$, the following holds:

$$
\begin{aligned}
&|L_r(f^1, \mathbf{y}) - L_r(f^2, \mathbf{y})| \\
&= \frac{1}{|Y^+||Y^-|} \sum_{p \in Y^+} \sum_{q \in Y^-} |\ell(f_p^1 - f_q^1) - \ell(f_p^2 - f_q^2)| \\
&\leq \frac{1}{|Y^+||Y^-|} \sum_{p \in Y^+} \sum_{q \in Y^-} \rho |f_p^1 - f_q^1 - f_p^2 + f_q^2| \qquad (\ell(u)\ is\ \rho - Lipschitz) \\
&\leq \rho \left[ \frac{1}{|Y^+||Y^-|} \sum_{p \in Y^+} \sum_{q \in Y^-} |f_p^1 - f_q^1 - f_p^2 + f_q^2|^2 \right]^{1/2} \qquad (Jense's\ Inequality) \\
&\leq \rho \left[ \frac{1}{|Y^+||Y^-|} \sum_{p \in Y^+} \sum_{q \in Y^-} \left\{ |f_p^1 - f_p^2|^2 + |f_q^1 - f_q^2|^2 \right\} \right]^{1/2} \qquad (|a-b|^2 \leq a^2 + b^2) \\
&= \rho \left[ \frac{1}{|Y^+||Y^-|} \left\{ |Y^-| \sum_{p \in Y^+} |f_p^1 - f_p^2|^2 + |Y^+| \sum_{q \in Y^-} |f_q^1 - f_q^2|^2 \right\} \right]^{1/2} \\
&\leq \rho \left[ \frac{\max\{|Y^+|, |Y^-|\}}{|Y^+||Y^-|} \sum_{j=1}^{c} |f_j^1 - f_j^2|^2 \right]^{1/2} \\
&= \frac{\rho}{\min\{|Y^+|, |Y^-|\}} \|f^1 - f^2\| \\
&\leq \rho \|f^1 - f^2\| \qquad\qquad (1 \leq \min\{|Y^+|, |Y^-|\} \leq \frac{c}{2}).
\end{aligned}
\tag{7}
$$

For the surrogate Subset Loss Eq.(4), $\forall f^1, f^2 \in \mathcal{F}$, the following holds:

$$
\begin{aligned}
&|L_s(f^1, \mathbf{y}) - L_s(f^2, \mathbf{y})| \\
&= |\max_{j \in [c]} \{\ell(\mathbf{y}_j f_j^1)\} - \max_{j \in [c]} \{\ell(\mathbf{y}_j f_j^2)\}| \\
&= |\ell(\mathbf{y}_q f_q^1) - \ell(\mathbf{y}_p f_p^2)| \qquad (w.l.o.g.\ assume\ q = \arg\max_{j \in [c]} \{\ell(\mathbf{y}_j f_j^1)\},\ p = \arg\max_{j \in [c]} \{\ell(\mathbf{y}_j f_j^2)\}) \\
&\leq |\ell(\mathbf{y}_q f_q^1) - \ell(\mathbf{y}_q f_q^2)| \qquad (w.l.o.g.\ assume\ \ell(\mathbf{y}_q f_q^1) \geq \ell(\mathbf{y}_p f_p^2).\ \ell(\mathbf{y}_p f_p^2) \geq \ell(\mathbf{y}_q f_q^2).) \\
&\leq \rho |\mathbf{y}_q f_q^1 - \mathbf{y}_q f_q^2| \qquad (\ell(u)\ is\ \rho - Lipschitz) \\
&= \rho |f_q^1 - f_q^2| \\
&\leq \rho \|f^1 - f^2\|_{\max} \\
&\leq \rho \|f^1 - f^2\| \qquad (\|a\|_{max} \leq \|a\|_2).
\end{aligned}
\tag{8}
$$

Furthermore, since the base loss function $\ell(u)$ is bounded by $B$, it's easy to verify these three surrogate loss functions are all bounded by $B$. $\quad\square$

### A.3 The base theorem for generalization analysis

Here, we give the base theorem used in the subsequent analysis, as follows.

**Theorem 1** (The base theorem for generalization analysis). *Assume the loss function $L : \mathbb{R}^c \times \{-1, +1\}^c \to \mathbb{R}_+$ is $\mu$-Lipschitz continuous w.r.t. the first argument and bounded by $M$. Besides, (1) and (2) in* Assumption 1 *are satisfied. Then, for any $\delta > 0$, with probability at least $1 - \delta$ over the draw of an i.i.d. sample $S$ of size $n$, the following generalization bound holds for all $f \in \mathcal{F}$:*

$$R(f) \le \hat{R}_S(f) + 2\sqrt{2}\mu\sqrt{\frac{c\Lambda^2 r^2}{n}} + 3M\sqrt{\frac{\log\frac{2}{\delta}}{2n}}. \tag{9}$$

*Proof.* It is straightforward to get this theorem by applying Theorem A.2 and Lemma A.2. $\quad\square$

## B   Learning guarantees between Hamming and Subset Loss

### B.1 The relationship between Hamming and Subset Loss

**Lemma 2** (The relationship between Hamming and Subset Loss). *For the classifier[1] $H(\mathbf{x}) = sgn \circ f(\mathbf{x})$, the following inequalities hold:*

$$L_h^{0/1}(H(\mathbf{x}), \mathbf{y}) \le L_s^{0/1}(H(\mathbf{x}), \mathbf{y}) \le L_s(f(\mathbf{x}), \mathbf{y}), \tag{10}$$

$$L_s^{0/1}(H(\mathbf{x}), \mathbf{y}) \le c L_h^{0/1}(H(\mathbf{x}), \mathbf{y}) \le c L_h(f(\mathbf{x}), \mathbf{y}). \tag{11}$$

*Proof.* For simplicity, we set $a_j = [\![ sgn(f_j(\mathbf{x}_i)) \ne \mathbf{y}_j ]\!] \in \{0, 1\}$, $j \in [n]$. Then,

$$\begin{aligned} L_h^{0/1}(H(\mathbf{x}), \mathbf{y}) &= \frac{1}{c}\{a_1 + \ldots + a_c\} = mean\{a_1, \ldots, a_c\} \\ L_s^{0/1}(H(\mathbf{x}), \mathbf{y}) &= \max\{a_1, \ldots, a_c\} \end{aligned} \tag{12}$$

Thus, it can be easily verified that

$$L_h^{0/1}(H(\mathbf{x}), \mathbf{y}) \le L_s^{0/1}(H(\mathbf{x}), \mathbf{y}), \ L_s^{0/1}(H(\mathbf{x}), \mathbf{y}) \le c L_h^{0/1}(H(\mathbf{x}), \mathbf{y}). \tag{13}$$

Besides, the following inequalities hold:

$$L_s^{0/1}(H(\mathbf{x}), \mathbf{y}) \le L_s(f(\mathbf{x}), \mathbf{y}), \ L_h^{0/1}(H(\mathbf{x}), \mathbf{y}) \le L_h(f(\mathbf{x}), \mathbf{y}). \tag{14}$$

$\quad\square$

### B.2 Learning guarantee of Algorithm $\mathcal{A}^h$ for Hamming Loss

**Theorem 2** (Optimize Hamming Loss, Hamming Loss bound). *Assume the loss function $L = L_h$, where $L_h$ is defined in Eq.(2). Besides,* Assumption 1 *is satisfied. Then, for any $\delta > 0$, with probability at least $1 - \delta$ over $S$, the following generalization bound in terms of* Hamming Loss *holds for all $f \in \mathcal{F}$:*

$$R_{0/1}^h(sgn \circ f) \le \hat{R}_S^h(f) + 2\sqrt{2}\rho\sqrt{\frac{\Lambda^2 r^2}{n}} + 3B\sqrt{\frac{\log\frac{2}{\delta}}{2n}}. \tag{15}$$

*Proof.* Since $L = L_h$, we can get its Lipschitz constant (i.e. $\frac{\rho}{\sqrt{c}}$) and bounded value (i.e. $B$) from Lemma 1. Then, applying Theorem 1 and the inequality $R_{0/1}^h(sgn \circ f) \le R^h(f)$, we can get this theorem. $\quad\square$

### B.3 Generalization bound for the classical Binary Relevance

**Corollary 1** (Binary Relevance [4], Hamming Loss bound). *Assume the loss function $L = L_h$, where $L_h$ is defined in Eq.(2) and the base loss function is the hinge loss $\ell(u) = \max(0, 1 - u)$. Besides, Assumption 1 is satisfied. Then, for any $\delta > 0$, with probability at least $1 - \delta$ over S, the following generalization bound in terms of Hamming Loss holds for all $f \in \mathcal{F}$.*

$$R_{0/1}^h(sgn \circ f) \le \hat{R}_S^h(f) + 2\sqrt{2}\sqrt{\frac{\Lambda^2 r^2}{n}} + 3B\sqrt{\frac{\log \frac{2}{\delta}}{2n}} \qquad (16)$$

*Proof.* Since the hinge loss $\ell(u) = \max(0, 1 - u)$ is 1-Lipschitz, we can straightforwardly get this corollary by applying Theorem 2. $\qquad \square$

### B.4 Learning guarantee of Algorithm $\mathcal{A}^h$ for Subset Loss

**Theorem 3** (Optimize Hamming Loss, Subset Loss bound). *Assume the loss function $L = cL_h$, where $L_h$ is defined in Eq.(2). Besides, Assumption 1 is satisfied. Then, for any $\delta > 0$, with probability at least $1 - \delta$ over S, the following generalization bound in terms of Subset Loss holds for all $f \in \mathcal{F}$:*

$$R_{0/1}^s(sgn \circ f) \le cR_{0/1}^h(sgn \circ f) \le c\hat{R}_S^h(f) + 2\sqrt{2}\rho c\sqrt{\frac{\Lambda^2 r^2}{n}} + 3Bc\sqrt{\frac{\log \frac{2}{\delta}}{2n}}. \qquad (17)$$

*Proof.* Since $L = cL_h$, we can get its Lipschitz constant (i.e. $\rho\sqrt{c}$) and bounded value (i.e. $Bc$) from Lemma 1. Then, applying Theorem 1, we can get that, for any $\delta > 0$, with probability at least $1 - \delta$ over S, the following generalization bound holds for all $f \in \mathcal{F}$:

$$cR^h(f) \le c\hat{R}_S^h(f) + 2\sqrt{2}\rho c\sqrt{\frac{\Lambda^2 r^2}{n}} + 3Bc\sqrt{\frac{\log \frac{2}{\delta}}{2n}}. \qquad (18)$$

Besides, from Lemma 2 (i.e. InEq.(11)), we can get the inequality $R_{0/1}^s(sgn \circ f) \le cR_{0/1}^h(sgn \circ f) \le cR^h(f)$. Thus, we can get this theorem. $\qquad \square$

### B.5 Learning guarantees of Algorithm $\mathcal{A}^s$ for Subset and Hamming Loss

**Theorem 4** (Optimize Subset Loss, Subset and Hamming Loss bounds). *Assume the loss function $L = L_s$, where $L_s$ is defined in Eq.(4). Besides, Assumption 1 is satisfied. Then, for any $\delta > 0$, with probability at least $1 - \delta$ over S, the following generalization bounds in terms of Subset and Hamming Loss hold for all $f \in \mathcal{F}$:*

$$R_{0/1}^h(sgn \circ f) \le R_{0/1}^s(sgn \circ f) \le \hat{R}_S^s(f) + 2\sqrt{2}\rho\sqrt{\frac{c\Lambda^2 r^2}{n}} + 3B\sqrt{\frac{\log \frac{2}{\delta}}{2n}}. \qquad (19)$$

*Proof.* Since $L = L_s$, we can get its Lipschitz constant (i.e. $\rho$) and bounded value (i.e. $B$) from Lemma 1. Then, applying Theorem 1, and the inequality $R_{0/1}^h(sgn \circ f) \le R_{0/1}^s(sgn \circ f) \le R^s(f)$ induced from Lemma 2 (i.e. InEq.(10)), we can get this theorem. $\qquad \square$

## C Learning guarantees between Hamming and Ranking Loss

### C.1 The relationship between Hamming and Ranking Loss

**Lemma 3** (The relationship between Hamming and Ranking Loss). *For the Hamming and Ranking Loss, the following inequality holds:*

$$L_r^{0/1}(f(\mathbf{x}), \mathbf{y}) \le cL_h^{0/1}(sgn \circ f(\mathbf{x}), \mathbf{y}) \le cL_h(f(\mathbf{x}), \mathbf{y}). \qquad (20)$$

*Further, if Assumption 2 is satisfied, the following inequality holds:*

$$L_h^{0/1}(t^* \circ f(\mathbf{x}), \mathbf{y}) \le cL_r^{0/1}(f(\mathbf{x}), \mathbf{y}) \le cL_r(f(\mathbf{x}), \mathbf{y}). \qquad (21)$$

*Proof.* (1). For the first inequality, the following holds:

$$L_r^{0/1}(f(\mathbf{x}), \mathbf{y}) \leq L_r^{0/1}(sgn \circ f(\mathbf{x}), \mathbf{y})$$

$$= \frac{1}{|Y^+||Y^-|} \sum_{p \in Y^+} \sum_{q \in Y^-} [\![sgn(f_p(\mathbf{x})) \leq sgn(f_q(\mathbf{x}))]\!]$$

$$= \frac{1}{|Y^+||Y^-|} \Bigg[ |Y^-| \sum_{p \in Y^+} [\![sgn(f_p(\mathbf{x}_i)) \neq 1]\!] + |Y^+| \sum_{q \in Y^-} [\![sgn(f_q(\mathbf{x}_i)) \neq -1]\!] -$$

$$\Bigg\{ \sum_{p \in Y^+} [\![sgn(f_p(\mathbf{x}_i)) \neq 1]\!] \Bigg\} \Bigg\{ \sum_{q \in Y^-} [\![sgn(f_q(\mathbf{x}_i)) \neq -1]\!] \Bigg\} \Bigg]$$

$$\leq \frac{1}{|Y^+||Y^-|} \Bigg[ |Y^-| \sum_{p \in Y^+} [\![sgn(f_p(\mathbf{x}_i)) \neq 1]\!] + |Y^+| \sum_{q \in Y^-} [\![sgn(f_q(\mathbf{x}_i)) \neq -1]\!] \Bigg]$$

$$= \frac{\sum_{p \in Y^+} [\![sgn(f_p(\mathbf{x}_i)) \neq 1]\!]}{|Y^+|} + \frac{\sum_{q \in Y^-} [\![sgn(f_q(\mathbf{x}_i)) \neq -1]\!]}{|Y^-|}$$

$$\leq \frac{|Y^+| + |Y^-|}{\min\{|Y^+|, |Y^-|\}} \frac{\sum_{p \in Y^+} [\![sgn(f_p(\mathbf{x}_i)) \neq 1]\!] + \sum_{q \in Y^-} [\![sgn(f_q(\mathbf{x}_i)) \neq -1]\!]}{|Y^+| + |Y^-|}$$

$$= \frac{c}{\min\{|Y^+|, |Y^-|\}} L_h^{0/1}(sgn \circ f(\mathbf{x}), \mathbf{y}) \qquad (|Y^+| + |Y^-| = c)$$

$$\leq \frac{c}{\min\{|Y^+|, |Y^-|\}} L_h(f(\mathbf{x}), \mathbf{y})$$

$$\leq c L_h(f(\mathbf{x}), \mathbf{y}) \qquad (1 \leq \min\{|Y^+|, |Y^-|\} \leq \frac{c}{2}).$$

$$(22)$$

In summary, the following holds:

$$L_r^{0/1}(f(\mathbf{x}), \mathbf{y}) \leq \frac{c}{\min\{|Y^+|, |Y^-|\}} L_h^{0/1}(sgn \circ f(\mathbf{x}), \mathbf{y}) \leq c L_h^{0/1}(sgn \circ f(\mathbf{x}), \mathbf{y}) \leq c L_h(f(\mathbf{x}), \mathbf{y}).$$

$$(23)$$

(2). For the second inequality, since the oracle optimal thresholding function $t^*(f)$ exists, the following holds:

$$L_h^{0/1}(t^* \circ f(\mathbf{x}), \mathbf{y}) = \frac{\sum_{p \in Y^+} [\![t^*(f_p(\mathbf{x}_i)) \neq 1]\!] + \sum_{q \in Y^-} [\![t^*(f_q(\mathbf{x}_i)) \neq -1]\!]}{|Y^+| + |Y^-|}$$

$$\leq \frac{\sum_{p \in Y^+} [\![t^*(f_p(\mathbf{x}_i)) \neq 1]\!]}{|Y^+|} + \frac{\sum_{q \in Y^-} [\![t^*(f_q(\mathbf{x}_i)) \neq -1]\!]}{|Y^-|}$$

$$= \frac{1}{|Y^+||Y^-|} \Bigg[ |Y^-| \sum_{p \in Y^+} [\![t^*(f_p(\mathbf{x}_i)) \neq 1]\!] + |Y^+| \sum_{q \in Y^-} [\![t^*(f_q(\mathbf{x}_i)) \neq -1]\!] \Bigg]$$

$$\leq \frac{1}{|Y^+||Y^-|} \Bigg[ |Y^-| \sum_{p \in Y^+} [\![t^*(f_p(\mathbf{x}_i)) \neq 1]\!] + |Y^+| \sum_{q \in Y^-} [\![t^*(f_q(\mathbf{x}_i)) \neq -1]\!]$$

$$+ \Bigg\{ \sum_{p \in Y^+} [\![t^*(f_p(\mathbf{x}_i)) \neq 1]\!] \Bigg\} \Bigg\{ \sum_{q \in Y^-} [\![t^*(f_q(\mathbf{x}_i)) \neq -1]\!] \Bigg\} \Bigg]$$

$$= \triangle.$$

$$(24)$$

In the following, we need to go on the proof by three cases.

Case (a). When $\sum_{p \in Y^+} [\![ t^*(f_p(\mathbf{x}_i)) \neq 1 ]\!] = 0$, the following holds:

$$\triangle = \frac{1}{|Y^+||Y^-|} \left[ |Y^+| \sum_{q \in Y^-} [\![ t^*(f_q(\mathbf{x}_i)) \neq -1 ]\!] \right]$$

$$\overset{①}{\leq} |Y^+| L_r^{0/1}(f(\mathbf{x}), \mathbf{y})$$

$$\leq c L_r^{0/1}(f(\mathbf{x}), \mathbf{y})$$

$$\leq c L_r(f(\mathbf{x}), \mathbf{y}). \tag{25}$$

For the inequality ①, the last element in the predicted relevant label list (according to the non-ascending order) should be a real relevant label due to the optimality of $t^*(f)$. Thus, the minimum value of $L_r^{0/1}(f(\mathbf{x}), \mathbf{y})$ should be $\frac{\sum_{q \in Y^-} [\![ t^*(f_q(\mathbf{x}_i)) \neq -1 ]\!]}{|Y^+||Y^-|}$. Hence, the inequality ① holds.

Case (b). When $\sum_{q \in Y^+} [\![ t^*(f_q(\mathbf{x}_i)) \neq -1 ]\!] = 0$, the following holds:

$$\triangle = \frac{1}{|Y^+||Y^-|} \left[ |Y^-| \sum_{p \in Y^+} [\![ t^*(f_p(\mathbf{x}_i)) \neq 1 ]\!] \right]$$

$$\overset{②}{\leq} |Y^-| L_r^{0/1}(f(\mathbf{x}), \mathbf{y})$$

$$\leq c L_r^{0/1}(f(\mathbf{x}), \mathbf{y})$$

$$\leq c L_r(f(\mathbf{x}), \mathbf{y}). \tag{26}$$

For the inequality ②, the first element in the predicted irrelevant label list (according to the non-ascending order) should be a real irrelevant label due to the optimality of $t^*(f)$. Thus, the minimum value of $L_r^{0/1}(f(\mathbf{x}), \mathbf{y})$ should be $\frac{\sum_{p \in Y^+} [\![ t^*(f_p(\mathbf{x}_i)) \neq 1 ]\!]}{|Y^+||Y^-|}$. Hence, the inequality ② holds.

Case (c). When $\sum_{p \in Y^+} [\![ t^*(f_p(\mathbf{x}_i)) \neq 1 ]\!] \neq 0$ and $\sum_{q \in Y^+} [\![ t^*(f_q(\mathbf{x}_i)) \neq -1 ]\!] = 0$, the following holds. Besides, for notation clarity, we first set

$$\clubsuit = \frac{1}{|Y^+||Y^-|} \left[ \sum_{p \in Y^+} [\![ t^*(f_p(\mathbf{x}_i)) \neq 1 ]\!] + \sum_{q \in Y^-} [\![ t^*(f_q(\mathbf{x}_i)) \neq -1 ]\!] \right.$$

$$\left. + \left\{ \sum_{p \in Y^+} [\![ t^*(f_p(\mathbf{x}_i)) \neq 1 ]\!] \right\} \left\{ \sum_{q \in Y^-} [\![ t^*(f_q(\mathbf{x}_i)) \neq -1 ]\!] \right\} \right]. \tag{27}$$

Then, we have

$$\triangle \leq \max\{|Y^+|, |Y^-|\} \times \clubsuit$$

$$\overset{③}{\leq} c L_r^{0/1}(f(\mathbf{x}), \mathbf{y})$$

$$\leq c L_r(f(\mathbf{x}), \mathbf{y}). \tag{28}$$

In this case, due to the optimality of $t^*(f)$, the last element in the predicted relevant label list (according to the non-ascending order) should be a real relevant label and the first element in the predicted irrelevant label list should be a real irrelevant label. Then, for the inequality ③, the minimum value of $L_r^{0/1}(f(\mathbf{x}), \mathbf{y})$ is $\clubsuit$, where in the predicted relevant list, all the real relevant labels (except the last element) have bigger score than other real irrelevant labels, and in the predicted irrelevant set, all the real relevant labels have bigger score than other real irrelevant labels (except the first element). Thus the inequality holds.

In summary, the following always holds:

$$L_h^{0/1}(t^* \circ f(\mathbf{x}), \mathbf{y}) \leq \max\{|Y^+|, |Y^-|\} L_r^{0/1}(f(\mathbf{x}), \mathbf{y}) \leq c L_r^{0/1}(f(\mathbf{x}), \mathbf{y}) \leq c L_r(f(\mathbf{x}), \mathbf{y}). \tag{29}$$

$\square$

## C.2 Learning guarantee of Algorithm $\mathcal{A}^h$ for Ranking Loss

**Theorem 5** (Optimize Hamming Loss, Ranking Loss bound). *Assume the loss function $L = cL_h$, where $L_h$ is defined in Eq.(2). Besides,* Assumption 1 *is satisfied. Then, for any $\delta > 0$, with probability at least $1 - \delta$ over S, the following generalization bound in terms of* Ranking Loss *holds for all $f \in \mathcal{F}$:*

$$R_{0/1}^r(f) \leq cR_{0/1}^h(sgn \circ f) \leq c\hat{R}_S^h(f) + 2\sqrt{2}\rho c\sqrt{\frac{\Lambda^2 r^2}{n}} + 3Bc\sqrt{\frac{\log\frac{2}{\delta}}{2n}}. \tag{30}$$

*Proof.* Since $L = cL_h$, we can get its Lipschitz constant (i.e. $\rho\sqrt{c}$) and bounded value (i.e. $Bc$) from Lemma 1. Then, applying Theorem 1, and the inequality $R_{0/1}^r(f) \leq cR_{0/1}^h(sgn \circ f) \leq cR^h(f)$ induced from Lemma 3 (i.e. InEq.(20)), we can get this theorem. $\qquad\square$

## C.3 Learning guarantee of Algorithm $\mathcal{A}^r$ for Ranking Loss

**Theorem 6** (Optimize Ranking Loss, Ranking Loss bound). *Assume the loss function $L = L_r$, where $L_r$ is defined in Eq.(6). Besides,* Assumption 1 *is satisfied. Then, for any $\delta > 0$, with probability at least $1 - \delta$ over S, the following generalization bound in terms of* Ranking Loss *holds for all $f \in \mathcal{F}$:*

$$R_{0/1}^r(f) \leq \hat{R}_S^r(f) + 2\sqrt{2}\rho\sqrt{\frac{c\Lambda^2 r^2}{n}} + 3B\sqrt{\frac{\log\frac{2}{\delta}}{2n}}. \tag{31}$$

*Proof.* Since $L = L_r$, we can get its Lipschitz constant (i.e. $\rho$) and bounded value (i.e. $B$) from Lemma 1. Then, applying Theorem 1 and the inequality $R_{0/1}^r(f) \leq R^r(f)$, we can get this theorem. $\qquad\square$

## C.4 Generalization bound for the classical Rank-SVM

**Corollary 2** (Rank-SVM [5], Ranking Loss bound). *Assume the loss function $L = L_r$, where $L_r$ is defined in Eq.(6) and the base loss function is the hinge loss $\ell(u) = \max(0, 1 - u)$. Besides,* Assumption 1 *is satisfied. Then, for any $\delta > 0$, with probability at least $1 - \delta$ over S, the following generalization bound in terms of* Ranking Loss *holds for all $f \in \mathcal{F}$:*

$$R_{0/1}^r(f) \leq \hat{R}_S^r(f) + 2\sqrt{2}\sqrt{\frac{c\Lambda^2 r^2}{n}} + 3B\sqrt{\frac{\log\frac{2}{\delta}}{2n}}. \tag{32}$$

*Proof.* Since the hinge loss $\ell(u) = \max(0, 1 - u)$ is 1-Lipschitz, we can straightforwardly get this corollary by applying Theorem 6. $\qquad\square$

## C.5 Learning guarantee of Algorithm $\mathcal{A}^r$ for Hamming Loss

**Theorem 7** (Optimize Ranking Loss, Hamming Loss bound). *Assume the loss function $L = cL_r$, where $L_r$ is defined in Eq.(6). Besides,* Assumption 1 *and 2 are satisfied. Then, for any $\delta > 0$, with probability at least $1 - \delta$ over S, the following generalization bound in terms of* Hamming Loss *holds for all $f \in \mathcal{F}$:*

$$R_{0/1}^h(t^* \circ f) \leq cR_{0/1}^r(f) \leq c\hat{R}_S^r(f) + 2\sqrt{2}\rho\sqrt{\frac{c^3\Lambda^2 r^2}{n}} + 3Bc\sqrt{\frac{\log\frac{2}{\delta}}{2n}}. \tag{33}$$

*Proof.* Since $L = cL_r$, we can get its Lipschitz constant (i.e. $\rho c$) and bounded value (i.e. $Bc$) from Lemma 1. Then, applying Theorem 1, and the inequality $R_{0/1}^h(t^* \circ f) \leq cR_{0/1}^r(f) \leq cR^r(f)$ induced from Lemma 3 (i.e. InEq.(21)), we can get this theorem. $\qquad\square$

# D Learning guarantees between Subset and Ranking Loss

## D.1 The relationship between Subset and Ranking Loss

In this section, we first analyze the relationships between Subset and Ranking Loss. Then, we analyze the learning guarantee of $\mathcal{A}^s$ on the Ranking Loss measure. Last, we analyze the learning guarantee of $\mathcal{A}^r$ on the Subset Loss measure.

**Lemma 4** (The relationship between Subset and Ranking Loss). *For the Subset and Ranking Loss, the following inequality holds:*

$$L_r^{0/1}(f(\mathbf{x}), \mathbf{y}) \leq L_s^{0/1}(sgn \circ f(\mathbf{x}), \mathbf{y}) \leq L_s(f(\mathbf{x}), \mathbf{y}). \tag{34}$$

*Further, if* Assumption 2 *is satisfied, the following inequality holds[1]:*

$$L_s^{0/1}(t^* \circ f(\mathbf{x}), \mathbf{y}) \leq c^2 L_r^{0/1}(f(\mathbf{x}), \mathbf{y}) \leq c^2 L_r(f(\mathbf{x}), \mathbf{y}). \tag{35}$$

*Proof.* For the first inequality, the following holds:

$$
\begin{aligned}
L_r^{0/1}(f(\mathbf{x}), \mathbf{y}) &\leq L_r^{0/1}(sgn \circ f(\mathbf{x}), \mathbf{y}) \\
&\leq L_s^{0/1}(sgn \circ f(\mathbf{x}), \mathbf{y}) && (the \ property \ of \ subset \ and \ ranking \ loss) \\
&\leq L_s(f(\mathbf{x}), \mathbf{y}) && (surrogate \ loss \ upper \ bounds \ 0/1 \ loss).
\end{aligned}
\tag{36}
$$

For the second inequality, we can get it from Lemma 2 and 3, i.e.

$$
\begin{aligned}
L_s^{0/1}(t^* \circ f(\mathbf{x}), \mathbf{y}) &\leq c L_h^{0/1}(t^* \circ f(\mathbf{x}), \mathbf{y}) && (Lemma \ 2) \\
&\leq c^2 L_r^{0/1}(f(\mathbf{x}), \mathbf{y}) && (Lemma \ 3) \\
&\leq c^2 L_r(f(\mathbf{x}), \mathbf{y}) && (surrogate \ loss \ upper \ bounds \ 0/1 \ loss).
\end{aligned}
\tag{37}
$$

□

From this lemma, we can observe that when optimizing Subset Loss with its surrogate loss function, it also optimizes an upper bound for Ranking Loss. Similarly, when optimizing Ranking Loss with its surrogate, it also optimizes an upper bound for Subset Loss which depends on $O(c^2)$.

## D.2 Learning guarantee of Algorithm $\mathcal{A}^s$ for Ranking Loss

$\mathcal{A}^s$ has a learning guarantee w.r.t Ranking Loss as follows.

**Theorem 8** (Optimize Subset Loss, Ranking Loss bound). *Assume the loss function $L = L_s$, where $L_s$ is defined in Eq.(4). Besides,* Assumption 1 *is satisfied. Then, for any $\delta > 0$, with probability at least $1 - \delta$ over S, the following generalization bound in terms of* Ranking Loss *holds for all $f \in \mathcal{F}$.*

$$R_{0/1}^r(f) \leq R_{0/1}^s(sgn \circ f) \leq \hat{R}_S^s(f) + 2\sqrt{2}\rho\sqrt{\frac{c\Lambda^2 r^2}{n}} + 3B\sqrt{\frac{\log\frac{2}{\delta}}{2n}} \tag{38}$$

*Proof.* Since $L = L_s$, we can get its Lipschitz constant (i.e. $\rho$) and bounded value (i.e. $B$) from Lemma 1. Then, applying Theorem 1, and the inequality $R_{0/1}^r(f) \leq R_{0/1}^s(sgn \circ f) \leq R^s(f)$ induced from Lemma 4 (i.e. InEq.(34)), we can get this theorem. □

From this theorem, we can observe that $\mathcal{A}^s$ has a generalization bound w.r.t. Ranking Loss depending on $O(\sqrt{c})$. Intuitively, the learning guarantee of $\mathcal{A}^s$ for Ranking Loss comes from its learning guarantee for Subset Loss.

### D.3 Learning guarantee of Algorithm $\mathcal{A}^r$ for Subset Loss

$\mathcal{A}^r$ has a learning guarantee w.r.t Subset Loss as follows.

**Theorem 9** (Optimize Ranking Loss, Subset Loss bound). *Assume the loss function $L = c^2 L_r$, where $L_r$ is defined in Eq.(4). Besides,* Assumption 1 *and* 2 *are satisfied. Then, for any $\delta > 0$, with probability at least $1 - \delta$ over S, the following generalization bound in terms of* Subset Loss *holds for all $f \in \mathcal{F}$.*

$$R_{0/1}^s(t^* \circ f) \le c^2 R_{0/1}^r(f) \le c^2 \hat{R}_S^r(f) + 2\sqrt{2}\rho\sqrt{\frac{c^5 \Lambda^2 r^2}{n}} + 3Bc^2\sqrt{\frac{\log\frac{2}{\delta}}{2n}} \tag{39}$$

*Proof.* Since $L = c^2 L_r$, we can get its Lipschitz constant (i.e. $\rho c^2$) and bounded value (i.e. $Bc^2$) from Lemma 1. Then, applying Theorem 1, and the inequality $R_{0/1}^s(t^* \circ f) \le c^2 R_{0/1}^r(f) \le c^2 R^r(f)$ induced from Lemma 4 (i.e. InEq.(35)), we can get this theorem. $\qquad\square$

From this theorem, we can observe that $\mathcal{A}^r$ has a generalization bound for Subset Loss which depends on $O(c^2)$. When the label space is large, $\mathcal{A}^r$ would perform poorly for Subset Loss.

## E Experiments

### E.1 Optimization

For $\mathcal{A}^h$ and $\mathcal{A}^s$, they are both convex optimization problems, which lots of off-the-shelf optimization algorithms can be employed to solve. Here we utilize the recent stochastic algorithm SVRG-BB [6] to efficiently train the linear models. For clarity, $\mathcal{A}^h$ and $\mathcal{A}^s$ can both be denoted by $\min_{\mathbf{W}} \frac{1}{n}\sum_i^n g_i(\mathbf{W})$, where $g_i(\mathbf{W}) = L(\mathbf{W}^\top \mathbf{x}_i, \mathbf{y}_i) + \lambda\|\mathbf{W}\|^2$ and $L$ denotes $L_h$ or $L_s$. Furthermore, the detailed optimization algorithm is summarized in Algorithm 1.

---

**Algorithm 1** SVRG-BB to solve $\mathcal{A}^h$ (or $\mathcal{A}^s$)

---

**Input**: initial step size $\eta_0$, update frequency $m$
**Output**: $\mathbf{W}^* \in \mathbb{R}^{d \times c}$

1: Initialize $\widetilde{\mathbf{W}}_0$ as zero matrix
2: **for** $s = 0, 1, ...$ **do**
3:    $\mathbf{G}_s = \frac{1}{n}\sum_{i=1}^n \nabla g_i(\widetilde{\mathbf{W}}_s)$
4:    **if** $s > 0$ **then**
5:      $\eta_s = \frac{1}{d}\|\widetilde{\mathbf{W}}_s - \widetilde{\mathbf{W}}_{s-1}\|_F^2 / \text{Tr}((\widetilde{\mathbf{W}}_s - \widetilde{\mathbf{W}}_{s-1})^\top(\mathbf{G}_s - \mathbf{G}_{s-1}))$
6:    **end if**
7:    $\mathbf{W}_0 = \widetilde{\mathbf{W}}_s$
8:    **for** $t = 0, 1, ..., m-1$ **do**
9:      Randomly pick $i_t \in \{1, 2, ..., n\}$
10:      $\mathbf{W}_{t+1} = \mathbf{W}_t - \eta_s(\nabla g_{i_t}(\mathbf{W}_t) - \nabla g_{i_t}(\widetilde{\mathbf{W}}_s) + \mathbf{G}_s)$
11:    **end for**
12:    $\widetilde{\mathbf{W}}_{s+1} = \mathbf{W}_m$
13: **end for**
14: **return** $\mathbf{W}^* = \widetilde{\mathbf{W}}_{s+1}$

---

## Footnotes

[1]Note that, for other classifiers, e.g. $H(\mathbf{x}) = t \circ f(\mathbf{x})$, the first inequalities (i.e. $L_h^{0/1}(H(\mathbf{x}), \mathbf{y}) \le L_s^{0/1}(H(\mathbf{x}), \mathbf{y})$ and $L_s^{0/1}(H(\mathbf{x}), \mathbf{y}) \le c L_h^{0/1}(H(\mathbf{x}), \mathbf{y})$) in the following also hold.

[1]Note that, this inequality depends on $O(c^2)$ and we believe it can be improved.