[Reviews · NeurIPS 2020]

Review 1

Summary and Contributions: This paper deals with theoretical issues in multi-label classification. Through Rademacher complexity and a vector contraction inequality, it derives and discusses generalisation error bounds for Hamming Loss, Subset Accuracy, and Ranking Loss in the case of Subset Loss optimisation.

Strengths: This paper provides appears to explain why, in the case of small label space, HL optimisation leads to better performances than SA.

Weaknesses: The relevance of the described results for the learning community at large is quite limited.

Correctness: This paper is mathematically sound. The theoretical findings are corroborated by experiments.

Clarity: This paper is very well written.

Relation to Prior Work: Definitely

Reproducibility: Yes

Additional Feedback: N.A.


Review 2

Summary and Contributions: The main idea is to use the Lipschitz continuity of loss functions with respect to the l2 norm (since the loss function takes a vector input) to study generalization bounds for multi-label learning.

Strengths: -

Weaknesses: This idea is well-known in the setting of multi-class classification, e.g., [1], [2]. Although this paper considers a different learning setting, i.e., multi-label learning, the theoretical analysis is exactly the same. In my opinion, the paper trivially extends the known results of multi-class classification to multi-label learning. The authors do not introduce any new technique and do not derive any surprising results. [1] Lei Y, Dogan Ü, Zhou DX, Kloft M. Data-dependent generalization bounds for multi-class classification. IEEE Transactions on Information Theory. 2019 Jan 21;65(5):2995-3021. [2] A. Maurer, “A vector-contraction inequality for rademacher complexities. International Conference on Algorithmic Learning Theory. Springer, 2016, pp. 3–17. ---- I slightly upgraded my evaluation; my concern persits: the theoretical analysis is straight forward.

Correctness: Yes.

Clarity: Yes.

Relation to Prior Work: Yes.

Reproducibility: No

Additional Feedback: -


Review 3

Summary and Contributions: The paper discusses generalization bounds for multi-label classification, using three common loss functions: Hamming loss, rank loss and subset zero-one loss. As main contribution, the authors claim that optimizing Hamming loss can be a good alternative for optimizing subset zero-one loss. This is also illustrated in a small experimental study using classical multi-label benchmarks.

Strengths: - the work is novel - the claims are based on theoretical results

Weaknesses: - some very restrictive assumptions are made - hence, the conclusions are not fully supported by the results

Correctness: The derivations are correct.

Clarity: Yes, it is very well written.

Relation to Prior Work: Yes, it is.

Reproducibility: Yes

Additional Feedback: The paper is very well written. Even though it is theoretical in nature, it is quite accessible for a broad audience. The performed analysis is novel, and the results are somewhat surprising. I do have concerns w.r.t. to the take-home message of the paper, because this message is contradicting the theoretical and empirical results obtained in [11]. In the introduction the authors mention the following to explain this contradiction “The analysis in [11] has inevitable limitations because the hypothesis space is uncontrained and the conditional distribution P(y|x) is known. These assumptions have a large gap with real learning algorithms where a constrained hypothesis space is used and P(y|x) is unknown”. It is not entirely clear to me what the authors want to say here, but perhaps they allude to the fact that in practice it is not so easy to estimate P(y|x). This is of course true, but at least one could try to estimate P(y|x) as good as possible. Several methods exist for doing this: structured SVMs, conditional random fields, probabilistic classifier chains, and deep versions of those methods. In contrast, the authors make a very strong assumption in Eqn. 7, by assuming a linear model per label, as in binary relevance. It has been shown in [11] that such a hypothesis is in general not capable of minimizing subset zero-one loss in a Bayes-optimal way. The hypothesis space in Eqn. 7 simply models P(y|x) as a product of marginals, thereby assuming conditional independence of labels. So, this hypothesis space is very restrictive. Under these restrictions, the results obtained by the authors are not so surprising. When assuming conditional independence between labels, the risk minimizers for Hamming loss and subset zero-one loss coincide, as shown in [11]. Nonetheless, the results of this paper are novel, and they are complementary to what already exists in the literature. I only have the feeling that the authors are not telling a very fair story. I think that this paper can be accepted, provided that the above comments are taken into account in a revised version. --------------------------------------------------------- Review update after author rebuttal: -------------------------------------------------------- There are indeed two competing frameworks for minimizing complex loss functions. Either you optimize a surrogate loss during training (known as direct utility maximization), or you fit a probabilistic model during training, followed by an inference phase for every test instance (known as the decision-theoretic approach). In the rebuttal the authors claim that their results are not comparable to [11], because they consider the first framework, whereas [11] considers the second framework. I don't agree. Perhaps it is a bit easier to analyze the difference between Hamming and subset zero-one loss in the decision-theoretic framework, but I don't see a reason why the conclusions should be different for the utility maximization framework. [11] has shown that two aspects are of crucial importance to minimize a complex loss such as the subset zero-one loss: 1. The algorithm should target that loss, e.g., by minimizing a surrogate of that loss. 2. The hypothesis space should be expressive enough. For the subset zero-one loss, [11] has shown that binary relevance leads to a hypothesis that is not expressive enough. I don't see why this would be different in the utility maximization framework. This could probably be easily illustrated in experiments on synthetic data, where complex dependencies between labels can be simulated. For example, a structured SVM that models dependencies between labels and minimizes subset zero-one loss is expected to outperform binary relevance on subset zero-one loss, while the latter might be better for Hamming loss. Thus, as stated in my initial review, the assumption made in Eqn. 7 leads to my opinion to a too restrictive hypothesis space, and hence wrong conclusions. I am not aware of any papers that prove my claim for subset zero-one loss in the utility maximization framework, but for the F-measure this has been illustrated in the work of Petterson and Caetano (NIPS 2010 and NIPS 2011).


Review 4

Summary and Contributions: For algorithms that optimise (surrogate) hamming, subset, and rank losses, the paper introduces (1) several inequalities associating the different (actual and surrogate) losses wrt a hypothesis (this allows to provide insight regarding the phenomenon of optimising for hamming loss achieving better subset-loss performance than directly optimising for subset loss) (2) generalisation bounds for each of the losses in terms of the others, further providing insights regarding their respective hardness and performance related to the different objectives, and (3) a set of experiments to validate the key applicable observations, namely that optimising for (surrogate) hamming (subset) loss performs better than the other when evaluating over hamming (subset) loss, respectively (with the observe exception mentioned in (1) above - only when the number of classes is small).

Strengths: The main contributions are the introduction of generalisation bounds for the different losses, and the inequalities between the different losses. The bounds are provided for kernel-based hypothesis classes, leverage recent Rademacher complexity results, and may serve to get similar bounds for other related losses and other hypothesis classes going forward.

Weaknesses: The proposed algorithms do not optimise the actual loss rather a surrogate, therefore the actual ordering among the optimised hypotheses depends on the tightness of the bounds and therefore any related conclusion should be qualified appropriately.

Correctness: The claims in the abstract (and in the rest of the paper) regarding the objective optimised by the algorithms (e.g., lines 6 and 7) should be qualified - the surrogate is optimised rather than the actual HL (or SA) loss.

Clarity: The paper is well written and easy to follow.

Relation to Prior Work: Yes

Reproducibility: Yes

Additional Feedback: Post rebuttal: After reading all reviews I decreased my score a notch. The key theoretical contribution/technique is not clear enough nor designated within the main body of the paper.

[Author Response · NeurIPS 2020]

We thank all reviewers for their helpful and constructive comments. We'll further improve in the final version. Below
we address their detailed comments.

**To R1**: Thanks for acknowledging our contributions.

**To R2**: We disagree on the judgement with our highest respect, due to the nontrivial technical differences and results.
In particular, our contributions are: (1) We introduce generalization bounds of learning algorithms on various losses, i.e.
HL, SA and HL. Besides, the inequalities between these (actual and surrogate) losses are introduced, which can help the
analysis extend to other forms of hypothesis classes; (2) based on the theoretical analysis, we explain the phenomenon
when in small label-space case, optimizing HL with its surrogate loss can have better performance on the SA measure
than directly optimizing SA with its surrogate loss; and (3) the experimental results support our theoretical analysis.

**Technique differences for bounds w.r.t. many measures:** The analysis techniques for multi-class classification
cannot be trivially extended to multi-label classification because we first need to analyze the relationships between these
measures. Besides, it's nontrivial to analyze the relationship between HL and RL, especially for the second inequality
(See Lemma 3 in Appendix C.1). Furthermore, as agreed by R4, our analysis can also be extended to other forms of
hypothesis classes (e.g. neural networks [15,19]), because the inequalities among these (actual and surrogate) losses are
independent of the hypothesis classes. More specifically, for multi-class classification, the performance of algorithms is
often evaluated in terms of only one measure (e.g. stand zero-one loss [*1]). Hence, the generalization bound analysis
of an algorithm is just provided for the measure [*1, *2, 17] that it aims to optimize. In comparison, for multi-label
classification, the performance of algorithms is evaluated in terms of many measures simultaneously, such as HL, SA,
and RL. Thus, this requires us to analyze the generalization bounds of an algorithm in terms of other measures in
addition to the measure that it aims to optimize. We'll add the discussions in the final version.

**Results:** With the above analysis techniques, we obtain new theoretical results that are substantially different from the
existing ones [11]. More specifically, [11] shows that SA and HL are conflicting measures — algorithms aiming to
optimize HL would perform poorly if evaluated on SA, and vice versa. In comparison, we show that when in small
label space case, optimizing HL with its surrogate loss can have better performance on the SA measure than directly
optimizing SA with its surrogate loss.

**To R3**: Thanks for acknowledging our novelty and sorry for the unclear parts. We'll make the comparison and statements
more precise in the final version. For the clarity of subsequent discussions, we distinguish two somewhat orthogonal
approach paradigms [*3] w.r.t. a loss $L^{0/1}$ for MLC: 1) one paradigm first estimates the conditional probability $P(\mathbf{y}|\mathbf{x})$
and gets the classifier by the optimal strategy w.r.t. $L^{0/1}$; 2) the other one directly optimizes $L^{0/1}$ with its surrogate loss
to find a classifier in a constrained parametric hypothesis space. In fact, the analysis in [11] is under the first paradigm,
while we are under the second one. Below, we discuss the pros and cons of each one in detail. We'll add the discussions
in the final version.

**Pros and cons of the analysis in [11]:** [11] can provide much insight for the first approach paradigm although there
is still a gap between the actual $P(\mathbf{y}|\mathbf{x})$ and its estimated one through many parametric methods (e.g., probabilistic
classifier chains, etc). In contrast, it may offer less insight for the second paradigm (e.g., binary relevance which directly
optimizes HL with its surrogate loss). More specifically, [11] assumes that the hypothesis space is unconstrained to
allow $P(\mathbf{y}|\mathbf{x})$ known, and gets the Bayes-optimal classifiers w.r.t. HL (i.e. $\mathbf{h}_H^*$) and SA (i.e. $\mathbf{h}_s^*$) by their corresponding
optimal strategy. Then, it analyzes the regret (a.k.a excess risk) upper bounds of $\mathbf{h}_H^*$ and $\mathbf{h}_s^*$ in terms of SA (i.e.,
Proposition 4) and HL (i.e., Proposition 5) respectively, and finds the bounds are large, which concludes that HL and
SA conflict with each other.

**Pros and cons of our analysis:** Our analysis can provide much insight for the second paradigm. In contrast, it may
offer less insight for the first paradigm. More specifically, we have no assumption of the conditional independence of
labels, and directly analyze the generalization bounds for the learning algorithms w.r.t. many measures. Although here
we consider the kernel-based hypothesis class, which includes the linear and non-linear model by specifying different
kernel functions, our analysis can be extended to other forms of hypothesis classes.

**To R4**: Thanks for acknowledging our contributions. Indeed, the learning algorithms optimize the loss with a surrogate
loss rather than the actual one. We'll make this clearer and qualify the related conclusions more appropriately in the
final version.

[*1] Lei et al. Multi-class svms: From tighter data-dependent generalization bounds to novel algorithms. NeurIPS 2015.

[*2] Lei et al. Data-dependent generalization bounds for multi-class classification. IEEE Trans. on Information Theory,
65(5):2995-3021, 2019.

[*3] Waegeman et al. On the bayes-optimality of f-measure maximizers. JMLR (15)3333-3388, 2014.


[Meta-Review · NeurIPS 2020]

This is an interesting theoretical paper that performs a cross-analysis of three popular loss functions used in multi-label classification. Despite the fact that some reviewers found the analysis straight-forward, as it is mainly based on known results either from multi-class classification or multi-label classification, the interpretation of the results from the multi-label classification perspective is very interesting. It is worth to underline that there is still a gap in theory for multi-label classification and this paper tries to fill it. Nevertheless, the paper has several flaws that makes the paper a borderline case. The current discussion about the frameworks used in [11] and in this submission is misleading (the rebuttal makes slightly better job in this regard). Both frameworks have their own limitations and those limitations should be clearly stated for both frameworks. The submission derives upper bounds, but these are only upper bounds, moreover for a specific, very constraint, model. This kind of analysis usually misses the aspect of consistency which is a central point of [11]. The results therein clearly indicate that minimization of both metrics can lead to completely different solutions having large regrets with respect to the other metric. This behavior was also confirmed empirically. The authors claim that "algorithms aiming to optimize HL often have better performance on the SA measure than the algorithms that optimize SA directly". Unfortunately, there is no reference given which would justify this claim. I have quickly checked several papers and have not observed that phenomenon. In turn, there are papers that claim the opposite: algorithms designed for SA perform well for HL (see, Classifier Chains: A Review and Perspectives, Read et al.). There are some results across different papers showing a very good performance of deep networks for both metrics while trained with different types of surrogate losses. These results, however, are not so surprising as [11] indicates specific situations under which optimization for both metrics coincides. One of them is low noise. So, if the model is enough expressive and well-trained, we can expect to obtain good results for both metrics. The authors do not discuss state-of-the-art methods used for optimizing SA. They are indeed usually very complex as they need to model conditional label dependencies to estimate (at least implicitly) the joint distribution of labels (since the Bayes decision for SA is the joint mode of the conditional distribution, not the marginal modes as for HL). The situation analyzed by authors is different. They use the same model for both metrics, but with a different loss optimized. Therefore, the authors should discuss the relation between the proposed algorithm and the algorithms modeling conditional label dependence, for example, Struct-SVMs. This algorithm enables incorporating label dependencies to the joint feature space defined on y and x and uses a loss function which is defined over all 2^c label vectors. It would be very interesting to learn what are the differences between the struct hinge loss and the hinge loss introduced by the authors. Is the specific surrogate loss used by the authors sufficient to find the joint mode of the distribution? Also the empirical study should be extended by a well-controlled experiment on synthetic data and comparison to other state-of-the-art methods. The paper also misses discussion about similar results for multi-class classification and structure-output prediction. For example, the authors should relate the bound obtained for SA to bounds for multi-class classification, since multi-label classification under SA can be meant as multi-class classification with an exponential number of classes. Moreover, the algorithm introduced by the authors could be used for multi-class classification as well. Assume that the original class labels are represented using binary codes. Then each bit of the code can be considered as a label in a multi-label problem. Solving this problem by the introduced algorithms would give a solution for the multi-class problem. I suppose that this is not a promising solution for multi-class classification. So, why it is good for multi-label classification?